# Rethinking Score Distilling Sampling for 3D Editing and Generation

**Xingyu Miao** [1]   **Haoran Duan** [2]   **Yang Long** [1]   **Jungong Han** [2]

## Abstract

Score Distillation Sampling (SDS) has emerged as a prominent method for text-to-3D generation by leveraging the strengths of 2D diffusion models. However, SDS is limited to generation tasks and lacks the capability to edit existing 3D assets. Conversely, variants of SDS that introduce editing capabilities often can not generate new 3D assets effectively. In this work, we observe that the processes of generation and editing within SDS and its variants have unified underlying gradient terms. Building on this insight, we propose Unified Distillation Sampling (UDS), a method that seamlessly integrates both the generation and editing of 3D assets. Essentially, UDS refines the gradient terms used in vanilla SDS methods, unifying them to support both tasks. Extensive experiments demonstrate that UDS not only outperforms baseline methods in generating 3D assets with richer details but also excels in editing tasks, thereby bridging the gap between 3D generation and editing. The code is available on: https://github.com/xingy038/UDS.

## 1. Introduction

In recent years, diffusion models (Hua et al., 2025) have achieved significant advancements across various fields (Saharia et al., 2022; Cao et al., 2024; Podell et al., 2023; Luo et al., 2023; Song & Ermon, 2019; Song et al., 2020b; Ho et al., 2020; Balaji et al., 2022). In particular, conditional diffusion models have been instrumental in enhancing conditional text generation, editing of 2D images (Hertz et al., 2022; Huberman-Spiegelglas et al., 2024; Lee et al., 2023), and audio processing (Ghosal et al., 2023; Huang et al., 2023). These models have demonstrated remarkable capabilities in numerous applications. However, the inherent complexity of 3D data (Miao et al., 2025) presents challenges for applying diffusion models in the 3D domain. Since most diffusion models are trained on 2D images, their effectiveness in generating 3D data is limited (Liu et al., 2024). Nevertheless, the extensive knowledge and generative priors derived from 2D image generation models provide valuable insights and potential applications for 3D data generation.

When working with 3D content, achieving realistic effects in generation and modification is crucial. Traditionally, these operations rely on specialized software and manual execution by experts. DreamFusion (Poole et al., 2022) introduced a breakthrough technique called Score Distillation Sampling (SDS) to overcome these limitations. By leveraging the generative priors of text-to-image diffusion models, SDS can generate 3D assets from text using Neural Radiance Fields (NeRF) (Mildenhall et al., 2021; Liu et al., 2020; Mildenhall et al., 2022) or 3D Gaussian Splatting (3D GS) (Kerbl et al., 2023). This approach surpasses the limitations of traditional 3D generation models and opens new possibilities for editing and generating 3D data.

However, previous studies have identified an averaging effect problem with SDS (Liang et al., 2023; Wang et al., 2024b; Wu et al., 2024; Lin et al., 2023; Wang et al., 2023). Specifically, the pseudo ground truths generated from different noise sources at the same viewing angle differ, and the update directions of these pseudo ground truths are applied to a single 3D model simultaneously. This results in the final output being overly smooth and lacking in detail. Additionally, SDS primarily focuses on generation and lacks editing capabilities. To address this issue, Delta Denoising Score (DDS) (Hertz et al., 2023) extended SDS to include editing capabilities but did not fully resolve the averaging effect problem. Although DDS performs well on 2D images, its effectiveness in 3D scenes is unsatisfactory. Posterior Distillation Sampling (PDS) (Koo et al., 2024) further investigates the reasons for DDS's poor performance in 3D scenes. The findings indicate that the lack of identity recognition in the gradient optimization term of DDS makes it difficult to retain information from the original scene, leading to editing failures.

In this work, we aim to develop a unified method for both 3D

[1] Department of Computer Science, Durham University, UK [2] Department of Automation, Tsinghua University, China. Correspondence to: Yang Long <yang.long@durham.ac.uk>, Haoran Duan <haoranduan@tsinghua.edu.cn>.

*Proceedings of the 42nd International Conference on Machine Learning*, Vancouver, Canada. PMLR 267, 2025. Copyright 2025 by the author(s).

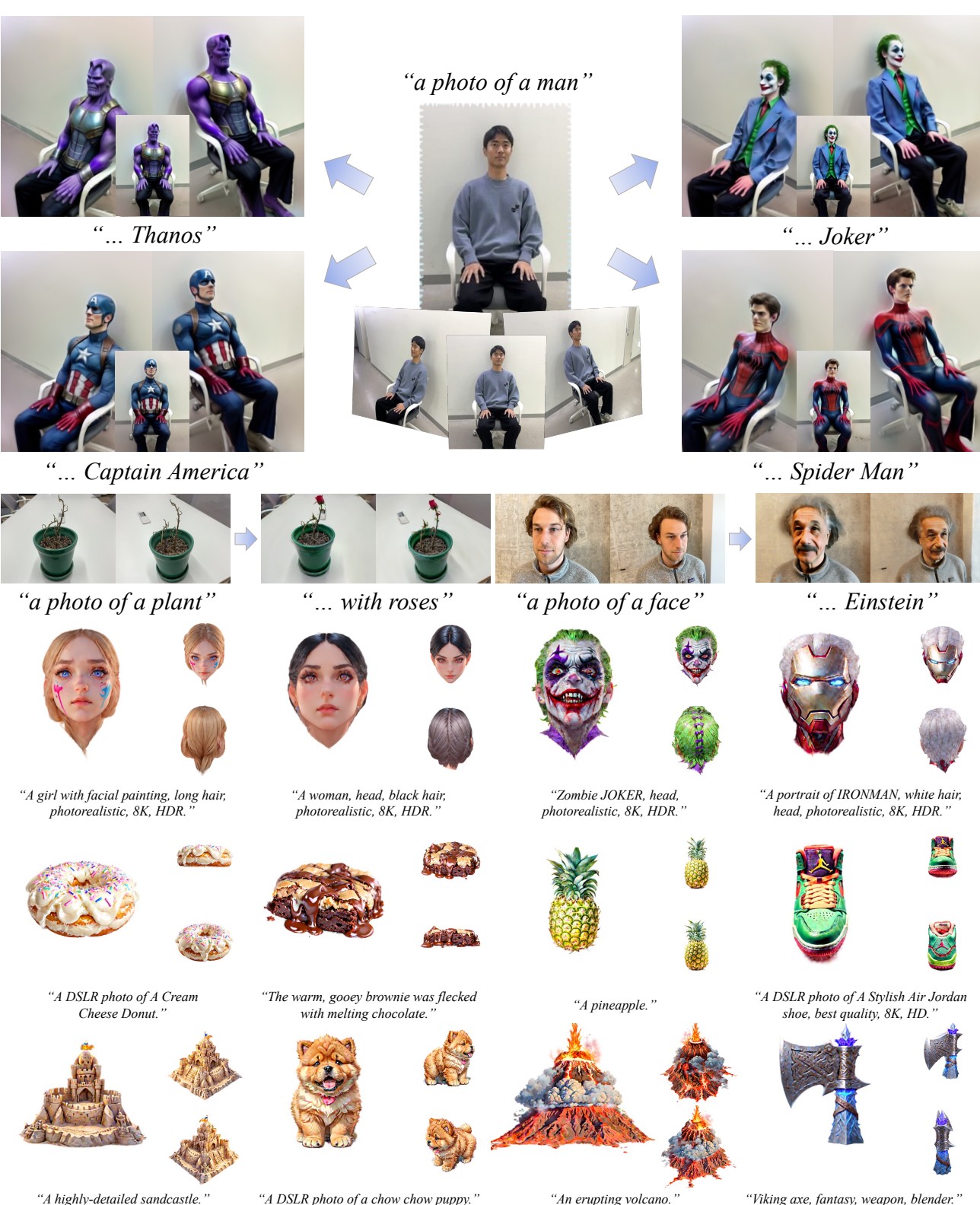

*Figure 1.* **Example of text-guided 3D editing and text-to-3D content generated from scratch by our UDS.** We achieve superior 3D editing and 3D generation results with photorealistic quality in a short training time. Please zoom in for details.

editing and generation. Specifically, we examine DDS and PDS—two representative 3D editing approaches—and disentangle their reconstruction term from the guidance term. This separation facilitates a more systematic analysis and comparison of various SDS variants. Then, we examine the factors contributing to their successes and failures, identifying notable similarities with the gradient terms used in recent DDIM-based SDS variants. Inspired by this insight, we propose Unified Distillation Sampling (UDS), to provide a general method for both 3D editing and generation tasks. Our UDS first approximates a clear latent $x_0$ representation, which serves as the reconstruction term. This reconstruction term is then combined with classifier-free guidance to supply the necessary gradient terms. UDS shows lower gradient variability and improved stability relative to previous methods, enabling the generation of superior edited results. In summary, our contributions are:

- We investigate text-to-3D generation and editing methods based on score sampling, identifying significant commonalities in their gradient optimization processes. We show that while these gradient terms serve different functions across various tasks, their forms remain consistent (Section 4.1). This consistency makes it possible to establish common methods across different tasks.

- We propose a novel Unified Distillation Sampling (UDS) that enables unified processing of text-guided 3D editing and text-to-3D generation (Section 4.2). UDS achieves improved editing and generation results by utilizing a single gradient formula to generate more stable gradients.

- Extensive experiments demonstrate the effectiveness of our UDS method across multiple applications, including the editing and generation of NeRF, the generation of 3D GS, and the editing of SVG. These results validate the effectiveness of UDS in achieving unified processing for both 3D editing and generation tasks.

## 2. Related Work

For generation tasks, Score Distillation Sampling (SDS) was initially introduced in DreamFusion (Poole et al., 2022) to directly optimize 3D representations using pre-trained 2D text-to-image diffusion models (Katzir et al., 2024). Score Jacobian Chaining (Wang et al., 2023) presents an alternative approach that achieves results comparable to SDS but is based on different mathematical principles. ProlificDreamer (Wang et al., 2024b) conducts a thorough examination of the SDS objective function and introduces a particle-based variational framework known as Variational Score Distillation (VSD), aimed at resolving issues of oversaturation, oversmoothing, and limited diversity inherent in SDS. Further-

more, Consistent3D (Wu et al., 2024) approaches SDS from the perspective of ordinary differential equations (ODE), developing a technique called Consistency Distillation Sampling (CSD) to mitigate over-smoothing and inconsistency. Similarly, LucidDreamer (Liang et al., 2023) explores the loss function of SDS and introduces Interval Score Matching (ISM), a method conceptually similar to CSD but utilizing reversible diffusion trajectories of DDIM (Song et al., 2020a; Zhuo et al., 2024).

In terms of editing, Haque et al. (Hertz et al., 2023) introduced Iterative Dataset Updating (IDU), a text-driven method for editing NeRF using Instruct-Pix2Pix (Brooks et al., 2023) or editing 3D GS (Wang et al., 2024a; Qu et al., 2025; Xu et al., 2024). This approach progressively substitutes original reference images with edited versions during NeRF reconstruction, gradually morphing the scene towards the edited state through adjustments in reconstruction loss or attention weights (Duan et al., 2023). Meanwhile, Mirzae et al. refined this process in Instruct-NeRF2NeRF (IN2N) (Haque et al., 2023) by targeting specific local areas for editing. Nevertheless, this iterative image replacement method struggles with edits requiring significant shifts across different views, such as complex geometric changes or the addition of new objects in undefined areas, and thus is mainly effective for appearance modifications. More recent approaches have abandoned the iterative IDU method in favor of directly applying SDS combined with segmentation techniques for NeRF editing (Li et al., 2023; Park et al., 2023; Zhuang et al., 2023). Hertz et al. (Hertz et al., 2023) introduced Delta Denoising Score (DDS), an editable variant of SDS designed to reduce the noise gradient direction in SDS to better maintain the details of the original image. However, DDS does not sufficiently ensure the retention of identity information. While this limitation is less apparent in image editing, it becomes problematic in NeRF-based 3D scene editing, as NeRF-based or 3D GS methods tend to be sensitive to gradients, which may lead to significant deviations from the original content. Conversely, Koo et al. (Koo et al., 2024) propose Posterior Distillation Sampling (PDS), focusing on enhancing editability and maintaining identity in text-aligned editing by minimizing the random latent matching loss they introduced. However, PDS fails to adequately disentangle the identity preservation term from the classifier term and inherits the high CFG weight of SDS (i.e., CFG=100), leading to over-saturation and a lack of diversity in the results. In contrast, we conduct an in-depth analysis of the gradient term in PDS and successfully disentangle the identity preservation term from the classifier term.

## 3. Background

**Diffusion Models** The diffusion model comprises a forward process that progressively perturbs the initial data $x_0$ with

noise $\epsilon$ and a reverse process that incrementally denoises the noisy data. The forward process is defined as follows:

$$\boldsymbol{x}_t = \sqrt{\bar{\alpha}_t}\boldsymbol{x}_0 + \sqrt{1-\bar{\alpha}_t}\epsilon, \quad \epsilon \sim \mathcal{N}(0, \mathbf{I}), \qquad (1)$$

where $\boldsymbol{x}t$ is the noisy latent representation of $\boldsymbol{x}_0$ at timestep $t$, $\{\bar{\alpha}_t\}_{t=0}^{T}$ (with $\bar{\alpha}_0 = 1$ and $\bar{\alpha}_T = 0$) denotes a set of time steps indexing a strictly monotonically decreasing noise schedule, and $\mathcal{N}(0, \mathbf{I})$ represents the Gaussian distribution. In the reverse process, a diffusion model denoising network $\epsilon_\phi$, parameterized by $\phi$, and a sampler prediction score function are utilized. Typically, the network is trained using denoising score matching:

$$\min_\phi \mathcal{L}(\phi) = \mathbb{E}_{t,\epsilon}\left[\|\epsilon_\phi(\boldsymbol{x}_t, t) - \epsilon\|_2^2\right]. \qquad (2)$$

**Score Distillation Sampling (SDS)** As discussed in Section 2, SDS (Poole et al., 2022) is a pioneering method for text-to-3D generation. It achieves this by seeking modes for the conditional posterior prior in the DDPM (Ho et al., 2020) latent space. Specifically, noise is added to rendered images $x := \boldsymbol{g}(\theta, c)$, where $\boldsymbol{g}(\cdot)$ represents a NeRF or 3D GS model, $\theta$ denotes the parameters of the NeRF or 3D GS model $\boldsymbol{g}(\cdot)$, and $c$ is the camera parameter. The method then distills knowledge from a pre-trained diffusion model with rich 2D priors to train the NeRF or 3D GS model. The optimization objective is:

$$\min_\theta \mathcal{L}_{\text{SDS}}(\theta) := \mathbb{E}_{t,c}\left[\omega(t)\|\epsilon_\phi(\boldsymbol{x}_t, t, y) - \epsilon\|_2^2\right], \quad (3)$$

where $\omega(t)$ is a time-dependent weighting function, $\epsilon$ is the standard Gaussian noise serving as the ground truth denoising direction of $\boldsymbol{x}_t$ at timestep $t$, and $\epsilon_\phi(\boldsymbol{x}_t, t, y)$ is the predicted denoising score given the condition $y$. Ignoring the UNet Jacobian (Poole et al., 2022), the gradient of the SDS loss is:

$$\nabla_\theta \mathcal{L}_{\text{SDS}} = \mathbb{E}_{t,\epsilon,c}\left[\omega(t)\left(\epsilon_\phi(\boldsymbol{x}_t, t, y) - \epsilon\right)\frac{\partial \boldsymbol{g}(\theta,c)}{\partial\theta}\right]. \quad (4)$$

For simplicity, we denote $\delta_{\boldsymbol{x}_t} := \epsilon_\phi(\boldsymbol{x}_t, t, y) - \epsilon$. The SDS employs Classifier Free Guidance, thus $\delta_{\boldsymbol{x}_t}$ in Equation (4) can be further expanded as:

$$\delta_{\boldsymbol{x}_t}^{\text{SDS}} := \underbrace{\epsilon_\phi(\boldsymbol{x}_t, t, \emptyset) - \epsilon}_{\delta_{\boldsymbol{x}_t}^{\text{recon}}} + w\underbrace{(\epsilon_\phi(\boldsymbol{x}_t, t, y) - \epsilon_\phi(\boldsymbol{x}_t, t, \emptyset))}_{\delta_{\boldsymbol{x}_t}^{\text{cls}}}, \quad (5)$$

where $w$ is the weight of Classifier-Free Guidance (Ho & Salimans, 2022). The SDS loss can thus be divided into two components: the reconstruction term $\delta_{\boldsymbol{x}_t}^{\text{recon}}$ and the classifier-free guidance term $\delta_{\boldsymbol{x}_t}^{\text{cls}}$.

## 4. Methodology

### 4.1. Revisiting SDS Variant for Editing

**What makes Delta Denoising Score (DDS) fail?** Inspired by (Poole et al., 2022), Hertz et al. (Hertz et al., 2023)

conceptualized image editing as a distribution-matching optimization problem. They treat the perturbation noise distributions of the original and edited images as two separate distributions that need to be aligned. First, they extract information from a pre-trained diffusion model and use text conditions to guide the image toward a specific region within the noise distribution. Then, they determine the update direction by estimating the score difference between the target and source distributions and perform the update edit. This approach addresses the issue of unclear and blurry images caused by the noise gradients generated by SDS. This implicit editing operation does not require the use of masks, which are defined as:

$$\nabla_\theta \mathcal{L}_{\text{DDS}} = \mathbb{E}_{t,\epsilon_t}\left[\omega(t)\left(\epsilon_\phi(\boldsymbol{x}_t^{\text{tgt}}, y^{\text{tgt}}, t) - \epsilon_\phi(\boldsymbol{x}_t^{\text{src}}, y^{\text{src}}, t)\right)\frac{\partial \boldsymbol{x}_0^{\text{tgt}}}{\partial\theta}\right], \quad (6)$$

where $\boldsymbol{x}_t^{\text{tgt}}$ and $\boldsymbol{x}_t^{\text{src}}$ represent the latent noise of $\boldsymbol{x}_0^{\text{tgt}}$ and $\boldsymbol{x}_0^{\text{src}}$ at timestep $t$ and they share the same noise $\epsilon_t$. Although DDS extends SDS to allow editing, DDS actually lacks the identity item, so it is difficult to preserve the source identity. This loss of identity information causes DDS to commonly fail in 3D editing.

**What makes Posterior Distillation Sampling (PDS) work?** The DDS demonstrates promising editability for 2D content. However, it falls short for 3D editing, which demands stronger identity preservation than 2D. To address this problem, the PDS aims to achieve both conformity to the text and preservation of the source's identity using the stochastic generative process of DDPM. Specifically, PDS introduces stochastic latents, ensuring that the latent noise of the reference and target in the latent space match. This can be expressed as:

$$\mathcal{L}_{\text{PDS}}(\boldsymbol{x}_0^{\text{tgt}} = \boldsymbol{g}(\theta)) := \mathbb{E}_{t,\epsilon}\left[\|\boldsymbol{z}_t^{\text{tgt}}(\boldsymbol{x}_t^{\text{tgt}}, y^{\text{tgt}}) - \boldsymbol{z}_t^{\text{src}}(\boldsymbol{x}_t^{\text{src}}, y^{\text{src}})\|_2^2\right], \quad (7)$$

where $\boldsymbol{z}_t(\cdot)$ is the stochastic latents at timestep $t$. The stochastic latent $\boldsymbol{z}_t(\cdot)$ is including the structural details of $x_0$ and is calculated as:

$$\boldsymbol{z}_t(\boldsymbol{x}_t, y) = \frac{\boldsymbol{x}_{t-1} - \mu_\phi(\boldsymbol{x}_t, y)}{\sigma_t}, \quad (8)$$

where the $\sigma_t := \frac{1-\bar{\alpha}_{t-1}}{1-\alpha_t}\beta_t$, and the posterior mean presents $\mu_\phi(\boldsymbol{x}_t, y) = \partial(t)\hat{\boldsymbol{x}}_0 + \psi(t)\boldsymbol{x}_t$. Here, $\partial(t)$ and $\psi(t)$ are coefficients about timestep $t$. By ignoring the UNet Jacobian term as SDS, the gradient of $\mathcal{L}_{\text{PDS}}$ is represented as:

$$\nabla_\theta \mathcal{L}_{\text{PDS}} = \mathbb{E}_{t,\epsilon}\left[\omega(t)\left(\boldsymbol{z}_t^{\text{tgt}}(\boldsymbol{x}_t^{\text{tgt}}, y^{\text{tgt}}) - \boldsymbol{z}_t^{\text{src}}(\boldsymbol{x}_t^{\text{src}}, y^{\text{src}})\right)\frac{\partial \boldsymbol{x}_0^{\text{tgt}}}{\partial\theta}\right]$$
$$= \mathbb{E}_{t,\epsilon}\left[c_0(t)\underbrace{(\boldsymbol{x}_0^{\text{tgt}} - \boldsymbol{x}_0^{\text{src}})}_{\text{Identity preservation}} + c_1(t)\underbrace{(\epsilon_\phi(\boldsymbol{x}_t^{\text{tgt}}, y^{\text{tgt}}, t) - \epsilon_\phi(\boldsymbol{x}_t^{\text{src}}, y^{\text{src}}, t))}_{\mathcal{L}_{\text{DDS}}}\right)\frac{\partial \boldsymbol{x}_0^{\text{tgt}}}{\partial\theta}\right]. \quad (9)$$

Here, $c_0(t)$ and $c_1(t)$ are coefficients defined with respect to the timestep $t$. We can observe that Equation (9) can be divided into two terms: one representing identity preservation, and the other corresponding to the equivalence for $\mathcal{L}_{\text{DDS}}$.

Since the $\boldsymbol{x}_0$ is unknown in the diffusion process, PDS using Tweedie's formula (Chung et al., 2022) that leverages The expectation of the posterior distribution $p(\boldsymbol{x}_0|\boldsymbol{x}_t)$ to approximate $\boldsymbol{x}_0$. It can be expressed as:

$$\boldsymbol{x}_0 \approx \hat{\boldsymbol{x}}_0 = \mathbb{E}[\boldsymbol{x}_0|\boldsymbol{x}_t] = \tfrac{1}{\sqrt{\bar{\alpha}}}(\boldsymbol{x}_t - \sqrt{1-\bar{\alpha}}\epsilon_\theta(\boldsymbol{x}_t, t, y)), \quad (10)$$

where $\epsilon_\theta(\boldsymbol{x}_t, t, y)$ is the prediction of the condition diffusion model. The main idea of PDS is to edit from the stochastic latent, but in fact, as shown in the PDS optimization term after decomposition by Equation (10), PDS adds an additional identity information term. This is also the key to the success of PDS.

**Disentangle DDS and PDS** We can link Equation (10) and Equation (6) to Equation (5), we can find that DDS and PDS can be expressed as:

$$\delta_{\boldsymbol{x}_t}^{\text{DDS}} := \underbrace{\epsilon_\phi(\boldsymbol{x}_t^{\text{tgt}}, t) - \epsilon_\phi(\boldsymbol{x}_t^{\text{src}}, t)}_{\delta_{\boldsymbol{x}_t}^{\text{recon}}} + w(\delta_{\boldsymbol{x}_t^{\text{tgt}}}^{\text{cls}} - \delta_{\boldsymbol{x}_t^{\text{src}}}^{\text{cls}}), \quad (11)$$

$$\delta_{\boldsymbol{x}_t}^{\text{PDS}} := \underbrace{(\hat{\boldsymbol{x}}_0^{\text{tgt}} - \hat{\boldsymbol{x}}_0^{\text{src}})}_{\text{Identity preservation}} + \underbrace{\epsilon_\phi(\boldsymbol{x}_t^{\text{tgt}}, t) - \epsilon_\phi(\boldsymbol{x}_t^{\text{src}}, t)}_{\delta_{\boldsymbol{x}_t}^{\text{recon}}} + w(\delta_{\boldsymbol{x}_t^{\text{tgt}}}^{\text{cls}} - \delta_{\boldsymbol{x}_t^{\text{src}}}^{\text{cls}}). \quad (12)$$

According to Equation (11) and Equation (12), we can observe that in the editing task, the gradient optimization term can actually be decomposed into two parts: one is the reconstruction term, and the other is the classifier-free guidance term. PDS operates primarily on stochastic latent. To simplify the description, we do not specify the coefficients in Equation (12). From the previous analysis, we know that DDS does not work well in editing, mainly because of the lack of identity-preserving terms. On the contrary, PDS effectively edits by introducing identity-preserving terms, although its complexity complicates the analysis.

**Link to generation task** Inspired by recent work (Yu et al., 2023), we know that the weight $w$ determines which term is dominant. In generation tasks, if the classifier-free guidance term is dominant, the generation proceeds as expected; the same applies to editing tasks. This observation also explains why DDS and PDS set $w$ to 100. In addition, combined with the latest DDIM-based generation methods (Liang et al., 2023), we further discovered that these methods can be summarized as:

$$\delta_{\boldsymbol{x}_t} := \underbrace{\epsilon_\phi(\boldsymbol{x}_t, t, \emptyset) - \epsilon_\phi(\boldsymbol{x}_{t-c}, t-c, \emptyset)}_{\delta_{\boldsymbol{x}_t}^{\text{recon}}} + w(\delta_{\boldsymbol{x}_t}^{\text{cls}}), \quad (13)$$

where $c$ is a defined time step interval and $w$ is a typical value (e.g., 7.5). We can find that Equation (13), Equation (11), and Equation (12) are very similar. This finding inspired us to explore whether a more unified form can be developed to accommodate both editing and generation tasks.

---

**Algorithm 1** Unified Distillation Sampling

1: Initialization: gradience scale $w$, time interval $c$
2: **while** $\theta$ is not converged **do**
3:     Sample: $\boldsymbol{x}_0 = g(\theta, c), \boldsymbol{\epsilon} \sim \mathcal{N}(0, \mathbf{I}), t \sim \mathcal{U}(1, 1000)$
4:     **if** Editing **then**
5:         **for** $i = [\text{src}, \text{tgt}]$ **do**
6:             $\boldsymbol{x}_t = \sqrt{\bar{\alpha}_t}\boldsymbol{x}_0 + \sqrt{1 - \bar{\alpha}_t}\boldsymbol{\epsilon}$
7:             Predict $\boldsymbol{\epsilon}_\phi(\boldsymbol{x}_t, t, y))$ and $\boldsymbol{\epsilon}_\phi(\boldsymbol{x}_t, t, \emptyset))$
8:             Approx. $\hat{\boldsymbol{x}}_0(t)$ via Eq. (10) or Eq. (15)
9:             $\delta_{\boldsymbol{x}_t} = \hat{\boldsymbol{x}}_0^t + w(\boldsymbol{\epsilon}_\phi(\boldsymbol{x}_t, t, y) - \boldsymbol{\epsilon}_\phi(\boldsymbol{x}_t, t, \emptyset))$
10:         **end for**
11:         $\nabla_\theta \mathcal{L}_{\text{UDS}} = \omega(t)(\delta_{\boldsymbol{x}_t}^{\text{tgt}} - \delta_{\boldsymbol{x}_t}^{\text{src}})$
12:         update $\boldsymbol{x}_0^{\text{tgt}}$ with $\nabla_\theta \mathcal{L}_{\text{UDS}}$
13:     **else**
14:         $\boldsymbol{x}_t = \sqrt{\bar{\alpha}_t}\boldsymbol{x}_0 + \sqrt{1 - \bar{\alpha}_t}\boldsymbol{\epsilon}$
15:         Predict $\boldsymbol{\epsilon}_\phi(\boldsymbol{x}_t, t, y))$ and $\boldsymbol{\epsilon}_\phi(\boldsymbol{x}_t, t, \emptyset))$
16:         $\boldsymbol{x}_{t-c} = \sqrt{\bar{\alpha}_{t-c}}\boldsymbol{x}_0 + \sqrt{1 - \bar{\alpha}_{t-c}}\boldsymbol{\epsilon}$
17:         Predict $\boldsymbol{\epsilon}_\phi(\boldsymbol{x}_{t-c}, t - c, \emptyset))$
18:         Approx. $\hat{\boldsymbol{x}}_0^t$ and $\hat{\boldsymbol{x}}_0^{t-c}$ via Eq. (10) or Eq. (15)
19:         $\nabla_\theta \mathcal{L}_{\text{UDS}} = \omega(t)(\hat{\boldsymbol{x}}_0^t - \hat{\boldsymbol{x}}_0^{t-c}) + w(\boldsymbol{\epsilon}_\phi(\boldsymbol{x}_t, t, y) - \boldsymbol{\epsilon}_\phi(\boldsymbol{x}_t, t, \emptyset)))$
20:         update $\boldsymbol{x}_0$ with $\nabla_\theta \mathcal{L}_{\text{UDS}}$
21:     **end if**
22: **end while**

---

### 4.2. Unified Distillation Sampling (UDS)

Based on the above analysis, in order to meet the requirements of editing tasks, the preservation of identity terms is particularly important. Therefore, we first combine the identity preservation term and the strategy without classifier guidance to effectively control the editing process, which can be expressed as:

$$\delta_{\boldsymbol{x}_t}^{\text{edit}} = \hat{\boldsymbol{x}}_0^{\text{tgt}} - \hat{\boldsymbol{x}}_0^{\text{src}} + w(\delta_{\boldsymbol{x}_t^{\text{tgt}}}^{\text{cls}} - \delta_{\boldsymbol{x}_t^{\text{src}}}^{\text{cls}}). \quad (14)$$

Note that Equation (14) and Equation (12) are very similar, but Equation (12) is a simplified representation we derived and is not the true gradient term of PDS. In addition, $\boldsymbol{x}_0$ is unknown during the generation process, thus we can leverage Equation (10) to approximate $\boldsymbol{x}_0$. However, this approach approximates $\boldsymbol{x}_0$ through single-step denoising. According to some recent work, we know that this approximation is inaccurate and may affect the preservation of identity information in editing tasks. Fortunately, we can use the deterministic sampling process of DDIM to obtain $\boldsymbol{x}_0$, which is expressed as:

$$\boldsymbol{x}_{t-1} = \sqrt{\bar{\alpha}_{t-1}} \left( \frac{\boldsymbol{x}_t - \sqrt{1-\bar{\alpha}_t}\boldsymbol{\epsilon}_\phi(\boldsymbol{x}_t, t, \emptyset)}{\sqrt{\bar{\alpha}_t}} \right) \\ + \sqrt{1 - \bar{\alpha}_{t-1}}\boldsymbol{\epsilon}_\phi(\boldsymbol{x}_t, t, \emptyset), \quad (15)$$

by using Equation (15), we can get $\boldsymbol{x}_0$ in an iteration. While for generation tasks, we simply modify the Equation (14)

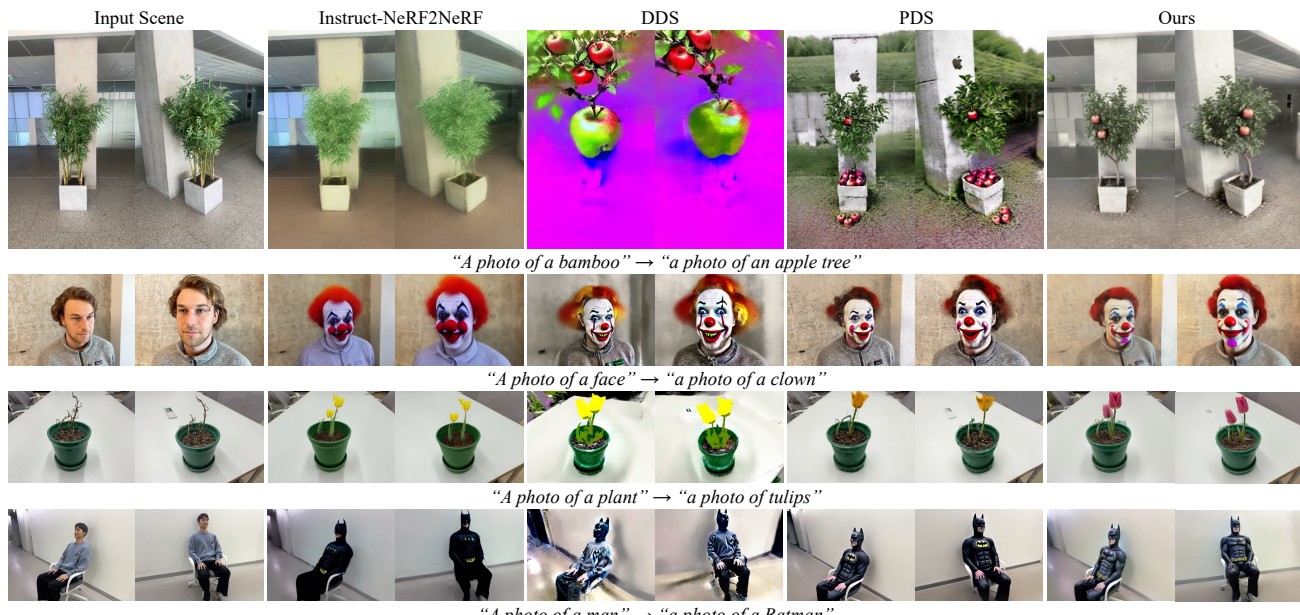

| Input Scene | Instruct-NeRF2NeRF | DDS | PDS | Ours |
|---|---|---|---|---|

*"A photo of a bamboo" → "a photo of an apple tree"*

*"A photo of a face" → "a photo of a clown"*

*"A photo of a plant" → "a photo of tulips"*

*"A photo of a man" → "a photo of a Batman"*

*Figure 2.* **Comparison with baseline methods in text-guided 3D editing.** We present visual editing results for both our UDS and the baseline methods. Experiments show that our UDS effectively edits 3D content to closely align with the input text prompts, while maintaining a high level of photorealism. Notably, DDS (Hertz et al., 2023) and PDS (Koo et al., 2024) set CFG is 100, while our is 7.5.

as:

$$\delta_{\boldsymbol{x}_t}^{\text{gen}} = \hat{\boldsymbol{x}}_0^t - \hat{\boldsymbol{x}}_0^{t-c} + w\delta_{\boldsymbol{x}_t}^{\text{cls}}, \tag{16}$$

where $c$ is a defined time step interval. Compared to Equation (13), we only modify the reconstruction term $\delta_{\boldsymbol{x}_t}^{\text{recon}}$ by replacing the predicted noise with the approximated $\hat{\boldsymbol{x}}_0$. Additional, insight from recent work (Yu et al., 2023; McAllister et al., 2024), we also can add a negative classifier-free guidance term to improve generation quality. The Equation (16) can be further rewrite as:

$$\delta_{\boldsymbol{x}_t}^{\text{gen}} = \hat{\boldsymbol{x}}_0^t - \hat{\boldsymbol{x}}_0^{t-c} + w(\delta_{\boldsymbol{x}_t}^{\text{cls}} - \delta_{\boldsymbol{x}_t}^{\text{neg}}), \tag{17}$$

Finally, we can express our $\delta_{\boldsymbol{x}_t}^{\text{UDS}}$ as:

$$\delta_{\boldsymbol{x}_t}^{\text{UDS}} = \Delta\hat{\boldsymbol{x}}_0 + w\Delta\delta_{\boldsymbol{x}_t}^{\text{cls}}. \tag{18}$$

The gradient of our $\mathcal{L}_{\text{UDS}}$ can be defined as follows:

$$\nabla_\theta \mathcal{L}_{\text{UDS}} = \mathbb{E}_{t,\boldsymbol{\epsilon},c}\left[\omega(t)(\Delta\hat{\boldsymbol{x}}_0 + w\Delta\delta_{\boldsymbol{x}_t}^{\text{cls}})\frac{\partial \boldsymbol{g}(\theta,c)}{\partial\theta}\right]. \tag{19}$$

The algorithm of our UDS is shown in the Algorithm 1.

### 4.3. Discuss with PDS and ISM

For the editing task, we derive the UDS in Appendix A.4. It can be observed that our final formulation is quite similar to that of PDS. When disregarding the predicted unconditional noises $\boldsymbol{\epsilon}_\phi(\boldsymbol{x}_t^{\text{tgt}}, t, \emptyset)$ and $\boldsymbol{\epsilon}_\phi(\boldsymbol{x}_t^{\text{src}}, t, \emptyset)$, PDS contains two time-dependent complex coefficients in its gradient terms,

whereas UDS exhibits a more streamlined formulation. The critical distinction is in the application of Tweedie's formula for $\boldsymbol{x}_0$ approximation: PDS employs conditional noise estimates while UDS utilizes unconditional counterparts. This fundamental difference explains UDS's capability to achieve comparable editing performance to PDS under small weights (e.g. 7.5) CFG settings. Essentially, UDS reallocates the weights between the reconstruction term and classifier-free guidance term through a simplified formulation, resulting in more intuitive gradient computation and significantly enhanced interpretability of the algorithm.

For the generation task, ISM adopts noise predictions that are highly correlated with the data at two time intervals as reconstruction terms. These reconstruction terms reflect the temporal change rate of noise predictions through gradients, providing directional guidance for the denoising process. In contrast, our approach directly utilizes the approximate $x_0$ to replace the reconstruction term by constructing it from the differences between $x_0$ predictions at two time intervals. Specifically, we use the difference between the predicted $x_0^{t-c}$ from the previous time step and the predicted $x_0^t$ from the current time step as the guiding signal for the reconstruction term. Since $x_0^{t-c}$ retains more detailed information in early predictions, while $x_0^t$ is more stable but may lose details in later predictions, this difference not only reflects the temporal change rate but also naturally introduces prior information for preserving details. In this way, we can provide directional guidance for the denoising process sim-

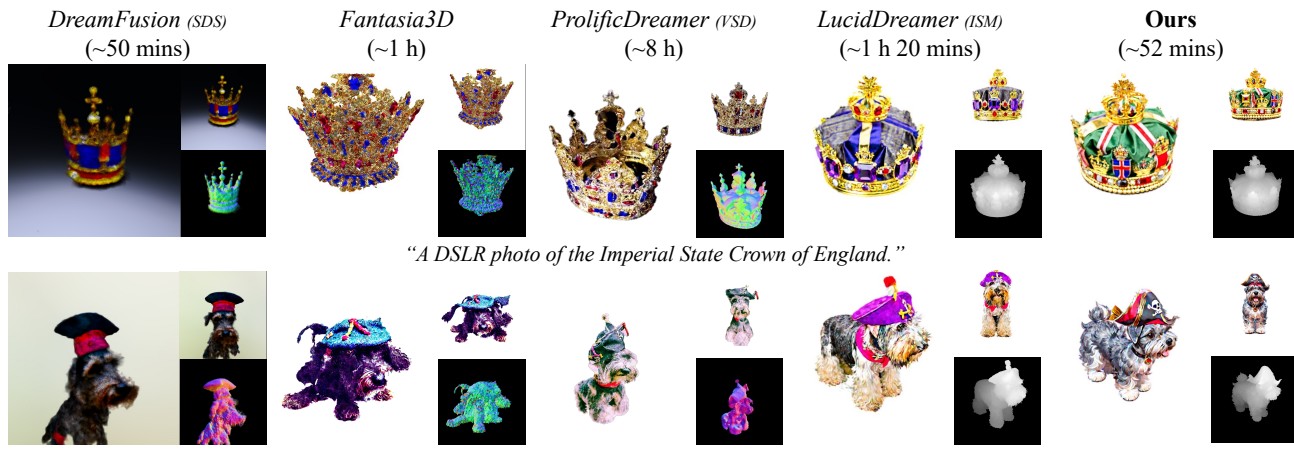

*DreamFusion (SDS)* (~50 mins)  *Fantasia3D* (~1 h)  *ProlificDreamer (VSD)* (~8 h)  *LucidDreamer (ISM)* (~1 h 20 mins)  **Ours** (~52 mins)

*"A DSLR photo of the Imperial State Crown of England."*

*"A DSLR photo of a Schnauzer wearing a pirate hat ."*

*Figure 3.* **Comparison with baseline methods in text-to-3D generation.** Experiments demonstrate that our approach can generate 3D content that closely aligns with the input text prompts, exhibiting high fidelity and intricate details. The running time of all methods is measured on a single 3090 GPU. Notably, we tried to reproduce ProlificDreamer and LucidDreamer, but failed to achieve the results in the LucidDreamer paper. Therefore, we directly use the visualization results in the LucidDreamer paper for these two methods.

| Methods | CLIP Score ↑ | User Preference Rate (%) ↑ |
|---|---|---|
| IN2N (Haque et al., 2023) | 0.2334 | 23.76 |
| DDS (Hertz et al., 2023) | 0.2030 | 4.52 |
| PDS (Koo et al., 2024) | 0.2395 | 30.20 |
| **Ours** | **0.2498** | **41.52** |

*Table 1.* **The quantitative comparison** of 3D editing performance between our method and others. Our approach quantitatively outperforms the baseline methods. **Bold** text indicates the best result.

| Methods | CLIP Score ↑ | User Preference Rate (%) ↑ |
|---|---|---|
| DreamFusion (Poole et al., 2022) | 0.2838 | 8.34 |
| Fantasia3D (Chen et al., 2023) | 0.2813 | 13.30 |
| ProlificDreamer (Wang et al., 2024b) | 0.2719 | 36.22 |
| **Ours** | **0.2962** | **42.14** |
| LucidDreamer(SDS) (Liang et al., 2023) | 0.2384 | 10.24 |
| LucidDreamer(ISM) (Liang et al., 2023) | 0.2897 | 41.39 |
| **Ours** | **0.2984** | **48.37** |

*Table 2.* **The quantitative comparison** of 3D generation between our method and others. Our approach quantitatively outperforms the baseline methods. **Bold** text indicates the best result.

ilar to ISM, while effectively preserving and transferring detailed information during denoising.

## 5. Experiments

### 5.1. Experiment Setup

For 3D editing, we implemented all methods using NeRFstudio and evaluated our approach, along with several baselines, on 8 scenes from real-world datasets provided by IN2N, using 37 pairs of source and target text prompts. We compared our method against three baseline approaches: IN2N (Haque et al., 2023), DDS (Hertz et al., 2023), and PDS (Koo et al., 2024). Since IN2N is based on IP2P, which specializes in

processing instruction-style text prompts, while DDS, PDS, and our method use the Stable Diffusion model (Saharia et al., 2022), which is designed to interpret description-style text prompts, we generate corresponding pairs of description and instruction-style prompts for evaluation. We conducted experiments using description-style prompts (e.g., "a photo of Batman") for DDS, PDS, and our method, while IN2N used instruction-style prompts (e.g., "Turn him into Batman"). We implemented NeRF-based methods using Threestudio (Guo et al., 2023) and 3D GS-based methods using LucidDreamer (Liang et al., 2023). Additionally, we compared several recent significant baseline methods, including DreamFusion (Poole et al., 2022), Fantasia3D (Chen et al., 2023), ProlificDreamer (Wang et al., 2024b), and LucidDreamer (Liang et al., 2023). For NeRF-based methods, we evaluated 15 prompts from Magic3D (Lin et al., 2023) and 415 prompts from DreamFusion (Poole et al., 2022). For 3D GS-based methods, where initialization significantly impacts the final results, we selected 43 reproducible prompts from recent works for evaluation.

### 5.2. Text Guided 3D Editing

**Results.** Figure 2 presents the qualitative comparisons of 3D editing with baseline methods. The editing results of the IN2N method are generally darker in color, and there are instances where editing is unsuccessful. For example, in row 1, the attempt to transform **bamboo** into an **apple tree** failed to exhibit any characteristics of an apple tree. Regarding DDS, all results display over-smoothing and over-saturation, and in row 1, the editing fails completely; it entirely loses the identity of the input scene, focusing solely on matching the input text. Although PDS achieves satisfactory results, it

DDIM Inverse        Tweedie's Formula

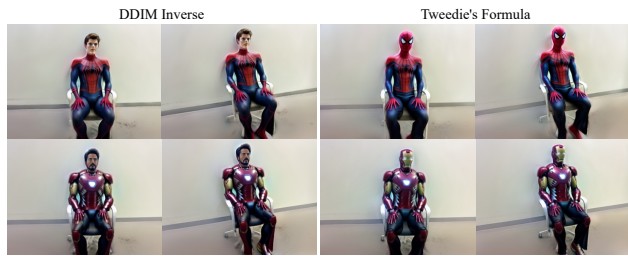

Figure 4. **Ablation** for different approximate methods.

also demonstrates instances of over-editing, such as in row 1 and row 2. In row 1, the Apple company's logo is inserted and the background is altered; in the second row, the clown's hair shape does not align with that of the original input. In contrast, our method makes appropriate modifications while best preserving the original identity information of the input scene. For instance, the clown's hair shape in the face remains unchanged.

To quantitatively evaluate our editing results, we measured the CLIP score (Radford et al., 2021), which assesses the similarity between the edited 2D renderings and the target text prompts in the CLIP embedding space. As shown in Table 1, our method outperforms the baselines in quantitative metrics. This is further confirmed by the qualitative results in Figure Figure 2, where other baseline methods struggle to produce clear textures, especially in scenes with complex details. To further assess the perceptual quality of the editing results, we conducted a user study comparing our method with the baselines. Following the setting used in PDS, users are shown the input 3D scene videos, the editing prompts, and the edited 3D scene videos produced by our method and the baselines. They are then asked to choose the most appropriate edited 3D scene video. As shown in Table 1, our editing results significantly outperform the baselines in human evaluation, receiving 41.25% of the selections compared to 30.20% for PDS, which was the second best.

**Ablation study.** For approximate the $x_0$, we can employ single denoise Tweedie's formula and multi-steps DDIM inverse processes. In the Figure 4, using a multi-step DDIM inversion process approximation better preserves identity information, such as facial features. In contrast, a single-step approximation with Tweedie's formula more accurately reflects the input text. However, the multi-step DDIM inversion process increases the optimization time.

### 5.3. Text-to-3D Generation

**Results.** Figure 3 present the qualitative comparisons of text-to-3D generation with baseline methods. We all use the stable diffusion 2.1 for distillation and all experiments are conducted on a 3090 GPU for fair comparison. Our method achieves high-fidelity and geometrically consistent results while requiring less time and fewer resources. The

DreamFusion         Fantasia3D

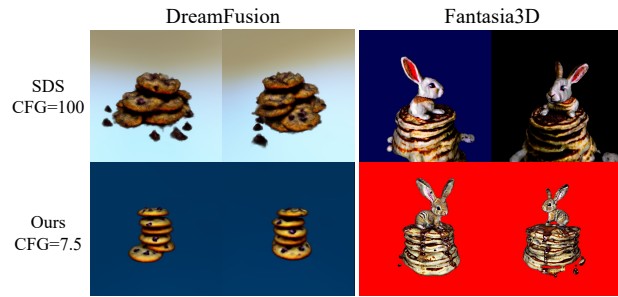

Figure 5. **Ablation** for SDS (Poole et al., 2022) and UDS with different generation frameworks.

W/o DDIM Inverse      W/ DDIM Inverse

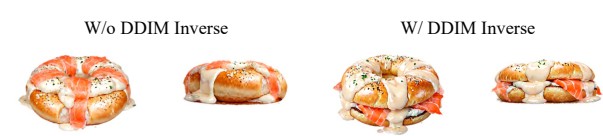

Figure 6. **Ablation for generation task.** Add noise by DDIM inverse strategy.

crown generated by our approach more closer to the text input, exhibiting a more precise geometric structure and realistic colors. Compared to the Schnauzers produced by other methods, the Schnauzer generated by ours features hair textures and an overall body shape that are closer to reality, with clearer and more detailed features.

As with the editing task, we use CLIP scores to quantitatively evaluate the generation results. In Table 2, our method outperforms the baseline in quantitative metrics. For fair comparison, we evaluate NeRF and 3D GS separately: we implement our method in the DreamerFusion framework to evaluate NeRF-based methods, and we implement our method in the LucidDreamer framework to evaluate 3D GS-based methods. To further evaluate the perceptual quality of the generated results, we also conduct a user study to compare our approach with baselines. In Table 2, our editing results are significantly better than the baselines in human evaluation, whether it is the NeRF-based method or the 3D GS-based method.

**Ablation study.** To verify the effectiveness of our method, we implemented UDS in DreamFusion and Fantasia3D frameworks respectively. In Figure 5, our method generates clearer and more realistic details compared to SDS. Additionally, in Figure 6, we perform ablations on DDIM reverse process noise addition. Our results show that while this approach improves the quality of generated results, it also increases time and resource costs.

## 6. Conclusion

In this work, we aim to explore unifying the SDS variant method into a comprehensive one applicable to both editing

and generation tasks. We investigate two SDS-based 3D scene editing, DDS and PDS, analyzing them to identify their successes and limitations. We observe remarkable similarities between these methods and the gradient terms used in recent DDIM-based SDS variants, although they play different roles in each task. Upon the SDS-based 3D editing method, we introduce UDS, an SDS variant method capable of both editing and generation objectives. Extensive experiments demonstrate the effectiveness of our method.

## Acknowledgements

This work was supported by the Royal Society International Exchanges Scheme-Towards Collaborative Cloud-Edge Deep Learning Deployment under Grant IEC/NSFC/223523; National Edge AI Hub for Real Data: Edge Intelligence for Cyber-disturbances and Data Quality EP/Y028813/1; and UK Medical Research Council (MRC) Innovation Fellowship under Grant MR/S003916/2.

## Impact Statement

This paper presents work whose goal is to advance the field of 3D Generation and Editing. There are many potential societal consequences of our work, none which we feel must be specifically highlighted here.

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

# A. Appendix

## A.1. Connection with other generation methods

We explore the connection between the proposed Unified Distillation Sampling (UDS) method and several other methods by implementing a 2D example (Figure 7). We understand that the role of the classifier-free guidance term is to continuously guide the sample to align with the text condition, while the role of the reconstruction term is to restore the sample to its initial state.

In SDS, the cosine similarity of the reconstruction term is low at the beginning of training and exhibits large fluctuations during training, while the classifier-free guidance term also fluctuates throughout the training process. In a high-dimensional mixed Gaussian distribution, if the optimization target is $\epsilon$, the maximum likelihood solution will appear at the origin. However, in reality, most of the probability density should be concentrated on a sphere with a radius of $\sqrt{d}$ centered at the origin. This indicates that in the early stages of training, when the sample lacks semantic information, the reconstruction term can effectively restore the sample. As training progresses and the sample gradually acquires semantic meaning, the reconstruction term begins to fluctuate due to mode-seeking behavior. The instability of the reconstruction term causes the classifier-free guidance term to also fluctuate, ultimately leading to issues such as oversmoothing. ISM, VSD, and our UDS methods not only avoid the loss of details and oversmoothing that may be caused by the reconstruction term by replacing $\epsilon$ in the reconstruction term but also allow for a smaller weight (e.g., 7.5) to be set for the classifier-free guidance term to prevent oversaturation.

Unlike SDS, the cosine similarity of the reconstruction term in ISM, VSD, and UDS is negative and fluctuates significantly at the beginning of training. As training progresses, the cosine similarity of the reconstruction term gradually approaches zero. We believe this is because, at the beginning of training, the rendered image is in an out-of-domain state, and the differences between different time steps are substantial. For VSD, the reconstruction term continuously aligns the distribution of pre-trained samples with the distribution of samples trained by LoRA (Hu et al., 2021), enhancing the detail performance of the samples. For UDS and ISM, the reconstruction term aligns the sample with its state at the previous timestep, which not only improves detail preservation but also ensures the coherence of the generation process. Additionally, compared with ISM, the reconstruction terms of UDS are aligned in an approximate latent space instead of directly on the noise. Consequently, the cosine similarity of the reconstruction terms in UDS approaches zero faster and exhibits smaller fluctuations. However, this also results in more significant fluctuations in the classifier-free guidance terms of UDS compared to ISM. Based on the generation

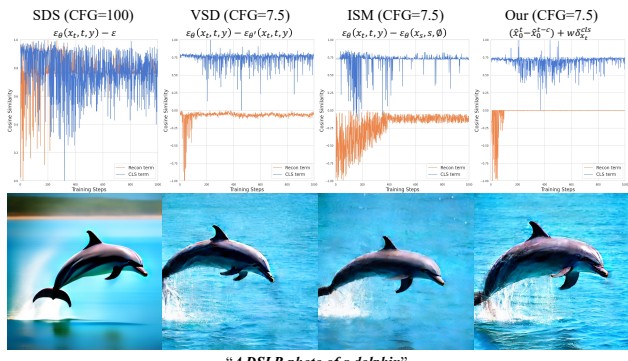

Figure 7. **Connection with other methods.** The cosine similarity between each item and the updated loss in the SDS (Poole et al., 2022), VSD (Wang et al., 2024b), ISM (Liang et al., 2023), and our UDS during the training process. The higher the similarity, the greater its weight in the loss.

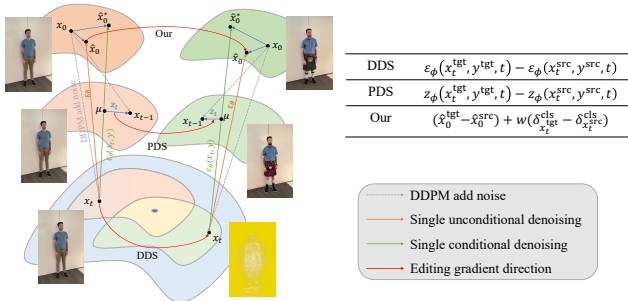

Figure 8. The workflow of 3D editing DDS (Hertz et al., 2023), PDS (Koo et al., 2024) and our UDS.

results, our method is closest to VSD, as evidenced by the similarity curve.

## A.2. Workflows

As illustrated in the Figure 8, we present schematic workflows of our UDS method alongside two baseline approaches, DDS (Hertz et al., 2023) and PDS (Koo et al., 2024). Specifically, our method computes gradients in a clean latent space distribution that is closer to the data distribution. In contrast, DDS calculates gradients on a pure noise distribution, while PDS performs gradient computations on a mixed distribution.

## A.3. More experiments

### A.3.1. TEXT GUIDED 3D EDITING

**Additional visualization results.** We present more editing results in Figure 15.

**Ablation for different $x_t$.** As presented in the main text, the PDS can preserve identity recognition because it in-

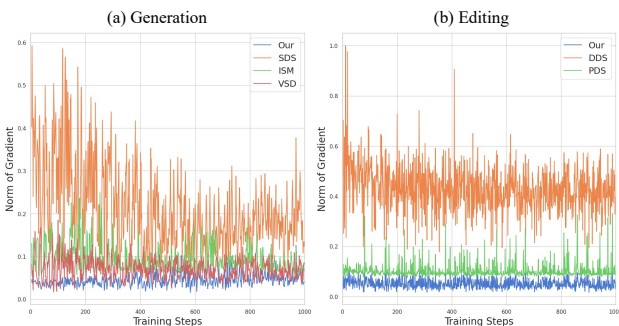

Figure 9. The normalized gradient norm for generated and edited tasks for 3D.

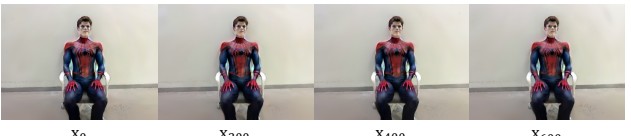

Figure 10. **Ablation for different** $x_t$**.** We use the inverse process of DDIM to denoise and obtain $x_t$ at different time steps for editing.

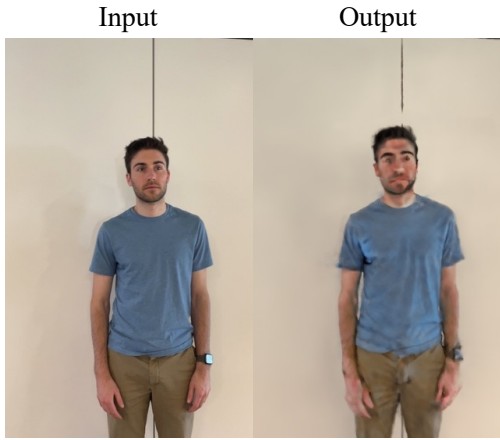

Figure 11. **Ablation for reference and target consistent prompt.** Even if the weight of the CFG is 0 the details of the image will change.

troduces an identity-preserving term of $x_0$ in the gradient, while our method directly combines this identity-preserving term with the classifier term as the gradient. Further observation of PDS reveals that it actually consists of the identity-preserving term and the noisy gradient term of DDS. When the coefficient is ignored, combining the identity term with the unconditional noise term is equivalent to adding the classifier-free guidance term to the latent space at a certain time step $t$ during the noise addition process of the diffusion model. We hypothesize whether it is possible to use the inverse process of DDIM to edit at any timestep $x_t$ obtained. We conducted an ablation experiment, as shown in Figure 10, and the results show that editing can be successfully performed at any time step. In other words, editing can be successful as long as it is not a combination of pure noise. Although no obvious changes are seen in Figure 10, what effect will the noise have on editing when $x_0$ is combined with noise? As shown in Figure 11, we used PDS to conduct an experiment in which the reference prompt is consistent with the target prompt, and the weight of CFG is set to 0, that is, the editing is completely dominated by $x_0$ and unconditional noise. In theory, the resulting image should not have any changes, but as shown in Figure 11, the man's face and watch have obviously changed. This shows that random noise affects the restoration of the image, causing some colors and textures to be abnormal. This does not have a big impact in the editing task, but it will affect the generated results in the generation task.

**The gradient analysis.** As shown in the right side of Figure 9, we show the normalized gradient norms generated by DDS, PDS, and our proposed UDS method. It can be seen that UDS and PDS are similar in the range of gradient norms. In contrast, the gradient change of the UDS method is more stable, while the gradient fluctuations of DDS and PDS are more obvious. This is mainly because DDS and PDS need to set CFG=100 to achieve convergence, which may lead to instability in the training process.

### A.3.2. TEXT-TO-3D GENERATION

**Additional visualization results.** We present more generation results in Figure 14.

**Ablation for different** $x_t$**.** We also investigate how the different $x_t$ affect the generation process. As shown in Figure 12, we observe that at higher timesteps, the colors of the generated 3D assets exhibit some abnormalities. For example, the sesame seeds and chopped green onions on the bagel gradually take on a blue. As discussed in Appendix A.3.1, this issue is caused by the higher timesteps introducing more random Gaussian noise, which may lead to such anomalies.

**The gradient analysis.** As shown on the left side of Figure 9, we present the normalized gradient norms generated by the SDS, VSD, ISM, and our proposed UDS methods. It is evident that UDS, ISM, and VSD with LoRA have similar gradient norm ranges. For UDS, ISM, and VSD, all set with CFG=7.5, UDS shows more stable gradient variations, while ISM and VSD exhibit more noticeable fluctuations. However, the fluctuations in VSD are smaller than those in ISM, as VSD uses LoRA to learn the distribution of 3D assets, whereas ISM experiences larger fluctuations due to additional accumulated errors introduced by noise dur-

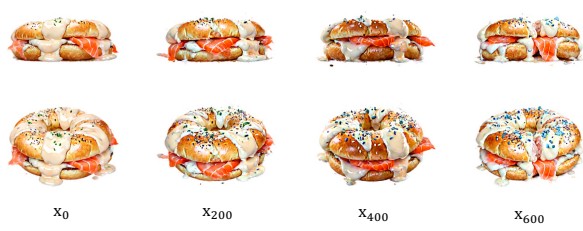

| | | | |
|---|---|---|---|
| x_0 | x_200 | x_400 | x_600 |

*Figure 12.* **Ablation for different** $x_t$**.** We use the inverse process of DDIM to denoise and obtain $x_t$ at different time steps for generation.

ing the reverse DDIM process, leading to instability in the reconstruction term, as discussed in Figure 7. While the differences in 2D results are minimal, these fluctuations cause inconsistencies in the local details of the generated 3D assets. Additionally, SDS with CFG=100 shows very large fluctuations. Although it eventually converges to a stable range, the range is quite large, leading to instability in the optimization process and poor 3D asset quality.

### A.3.3. TEXT GUIDED SVG EDITING

We also conduct experiments on some SVG images from VectorFusion (Jain et al., 2023).

**Results.** Figure 13 presents a detailed qualitative comparison of text-guided SVG editing across various baseline methods. For consistency, we utilize the stable diffusion 1.5 model for distillation in all approaches, ensuring that the comparisons are performed under fair conditions. All experiments were conducted on an NVIDIA 3090 GPU to maintain fairness in terms of computational resources. As shown in Figure 13, while all methods successfully modify the input SVG according to the target text prompts, our UDS and PDS methods demonstrate superior performance in preserving the structural semantics of the original SVG. This is particularly evident in maintaining the overall color scheme and structure of the input SVG, which is most prominent in the third row of Figure 13. The ability to retain these visual and semantic features distinguishes our methods from the baseline approaches. This qualitative advantage is further corroborated by the quantitative results. As indicated in Table 3, our method outperforms the baseline approaches by a significant margin in the LPIPS metric (Zhang et al., 2018), which is specifically designed to measure the perceptual similarity and fidelity to the input SVG. This suggests that our method maintains a higher level of detail and consistency with the original SVG compared to other methods. Despite the higher fidelity, our CLIP score remains competitive, showing that our approach balances the trade-off between preserving the integrity of the input and effectively implementing the changes dictated by the text prompt. In

| Methods | CLIP Score ↑ | LPIPS ↓ | User Preference Rate (%) ↑ |
|---|---|---|---|
| SDS (Poole et al., 2022) | 0.2507 | 0.4381 | 26.51 |
| DDS (Hertz et al., 2023) | 0.2370 | 0.5236 | 19.64 |
| PDS (Koo et al., 2024) | 0.2432 | 0.3803 | 26.21 |
| **Ours** | **0.2576** | **0.3489** | **27.64** |

*Table 3.* The quantitative comparison of SVG editing performance between our method and others. **Bold** text indicates the best result in each column.

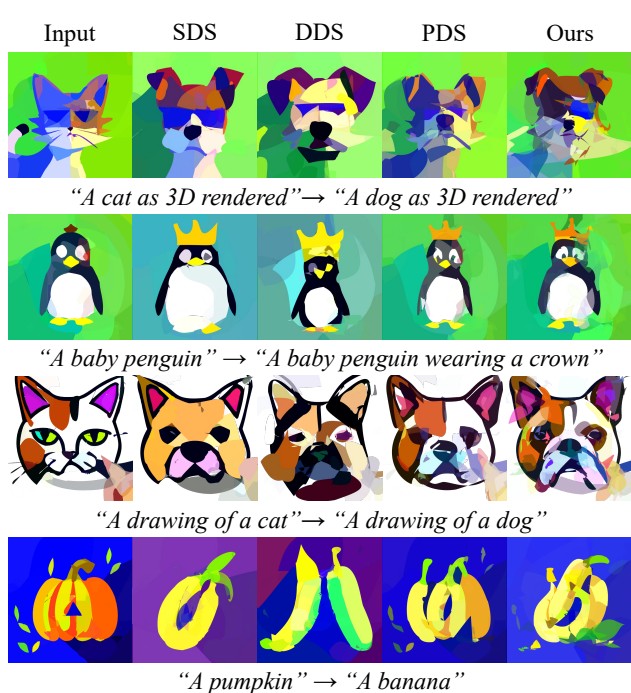

| Input | SDS | DDS | PDS | Ours |
|---|---|---|---|---|

*"A cat as 3D rendered"→ "A dog as 3D rendered"*

*"A baby penguin" → "A baby penguin wearing a crown"*

*"A drawing of a cat"→ "A drawing of a dog"*

*"A pumpkin" → "A banana"*

*Figure 13.* **Comparison with baseline methods in SVG editing.** We present visual editing results for other methods and ours. Our method preserves more obvious element information such as structure and color.

addition, we also conduct a user study, the results of which are summarized in Table 3. This user study followed the same setup as the one we employed for 3D editing, ensuring a consistent evaluation framework. The results of the user study show that, in terms of subjective human evaluation, our UDS method performs comparably to the baseline methods SDS and PDS. This parity in human preference highlights the effectiveness and reliability of our approach, both from a quantitative and qualitative perspective.

### A.4. Derivation of Unified Distillation Sampling

To derive the Unified Distillation Sampling (UDS) comprehensively, we begin by expressing the gradient of UDS as:

$$\nabla_\theta \mathcal{L}_{\text{UDS}} = \mathbb{E}_{t,\epsilon,c} \left[ \omega(t) \left( \hat{\boldsymbol{x}}_0^{\text{tgt}} - \hat{\boldsymbol{x}}_0^{\text{src}} + (\delta_{\boldsymbol{x}_t^{\text{tgt}}}^{\text{cls}} - \delta_{\boldsymbol{x}_t^{\text{src}}}^{\text{cls}}) \right) \frac{\partial \boldsymbol{g}(\theta,c)}{\partial \theta} \right]. \quad (20)$$

The term $\hat{\boldsymbol{x}}_0^{\text{tgt}} - \hat{\boldsymbol{x}}_0^{\text{src}} + (\delta_{\boldsymbol{x}_t^{\text{tgt}}}^{\text{cls}} - \delta_{\boldsymbol{x}_t^{\text{src}}}^{\text{cls}})$ can be decomposed as:

$$\tfrac{1}{\sqrt{\bar{\alpha}}} \left( \boldsymbol{x}_t^{\text{tgt}} - \sqrt{1-\bar{\alpha}} \, \epsilon_\phi \left( \boldsymbol{x}_t^{\text{tgt}}, t, \emptyset \right) \right) - \tfrac{1}{\sqrt{\bar{\alpha}}} \left( \boldsymbol{x}_t^{\text{src}} - \sqrt{1-\bar{\alpha}} \, \epsilon_\phi \left( \boldsymbol{x}_t^{\text{src}}, t, \emptyset \right) \right) \quad (21)$$

$$+ \epsilon_\phi \left( \boldsymbol{x}_t^{\text{tgt}}, y^{\text{tgt}}, t \right) - \epsilon_\phi \left( \boldsymbol{x}_t^{\text{src}}, y^{\text{src}}, t \right) \quad (22)$$

Subsequently, this expression simplifies to:

$$\tfrac{1}{\sqrt{\bar{\alpha}}} (\boldsymbol{x}_t^{\text{tgt}} - \boldsymbol{x}_t^{\text{src}}) - \tfrac{\sqrt{1-\bar{\alpha}}}{\sqrt{\bar{\alpha}}} (\epsilon_\phi(\boldsymbol{x}_t^{\text{tgt}}, t, \emptyset) - \epsilon_\phi(\boldsymbol{x}_t^{\text{src}}, t, \emptyset)) \quad (23)$$

$$+ \epsilon_\phi \left( \boldsymbol{x}_t^{\text{tgt}}, y^{\text{tgt}}, t \right) - \epsilon_\phi \left( \boldsymbol{x}_t^{\text{src}}, y^{\text{src}}, t \right) \quad (24)$$

Here, the latent noisy $\boldsymbol{x}_t$ is defined as:

$$\boldsymbol{x}_t = \sqrt{\bar{\alpha}_t} \boldsymbol{x}_0 + \sqrt{1-\bar{\alpha}_t} \epsilon \quad (25)$$

We introduce the following notations for simplification:

$$\hat{\epsilon}_t^{\text{src}} := \epsilon_\phi \left( \boldsymbol{x}_t^{\text{src}}, y^{\text{src}}, t \right) - \frac{\sqrt{1-\bar{\alpha}}}{\sqrt{\bar{\alpha}}} \epsilon_\phi(\boldsymbol{x}_t^{\text{src}}, t, \emptyset) \quad (26)$$

$$\hat{\epsilon}_t^{\text{tgt}} := \epsilon_\phi \left( \boldsymbol{x}_t^{\text{tgt}}, y^{\text{tgt}}, t \right) - \frac{\sqrt{1-\bar{\alpha}}}{\sqrt{\bar{\alpha}}} \epsilon_\phi(\boldsymbol{x}_t^{\text{tgt}}, t, \emptyset) \quad (27)$$

Thus, the gradient of the UDS loss function can be succinctly expressed as:

$$\nabla_\theta \mathcal{L}_{\text{UDS}} = \mathbb{E}_{t,\epsilon,c} \left[ \omega(t) \left( \boldsymbol{x}_0^{\text{tgt}} - \boldsymbol{x}_0^{\text{src}} + (\hat{\epsilon}_t^{\text{tgt}} - \hat{\epsilon}_t^{\text{src}}) \right) \frac{\partial \boldsymbol{g}(\theta,c)}{\partial \theta} \right]. \quad (28)$$

### A.5. User Study Details

We conducted a user study to evaluate the performance of different methods based on human preferences. In the generation task, we showed participants a side-by-side comparison of 3D assets generated by each method. In each trial, participants received a text prompt and a rotated video of multiple candidate 3D assets generated using different methods. In the editing task, we showed the side-by-side effect of each method editing a 3D scene. In each trial, participants received a text prompt, a reference scene, and videos of multiple candidate 3D scenes edited using different methods. We collected responses from a total of 102 participants. Each participant randomly performed 50 to 100 trials, and their selection data were recorded for subsequent analysis. To ensure the diversity and fairness of the evaluation results, the order of presentation of candidate content was randomly arranged in 50 to 100 trials for each participant.

### A.6. Failure Case

In generation tasks, success depends on the initialization; if the initialization is poor, the generation is likely to fail. In editing tasks, when there is a large gap between the prompt and the original scene, failure may occur. For example, when changing the prompt from "A photo of a plant" to "a photo of balls", the result often does not generate balls on the branches, but rather places them at random positions in the scene.

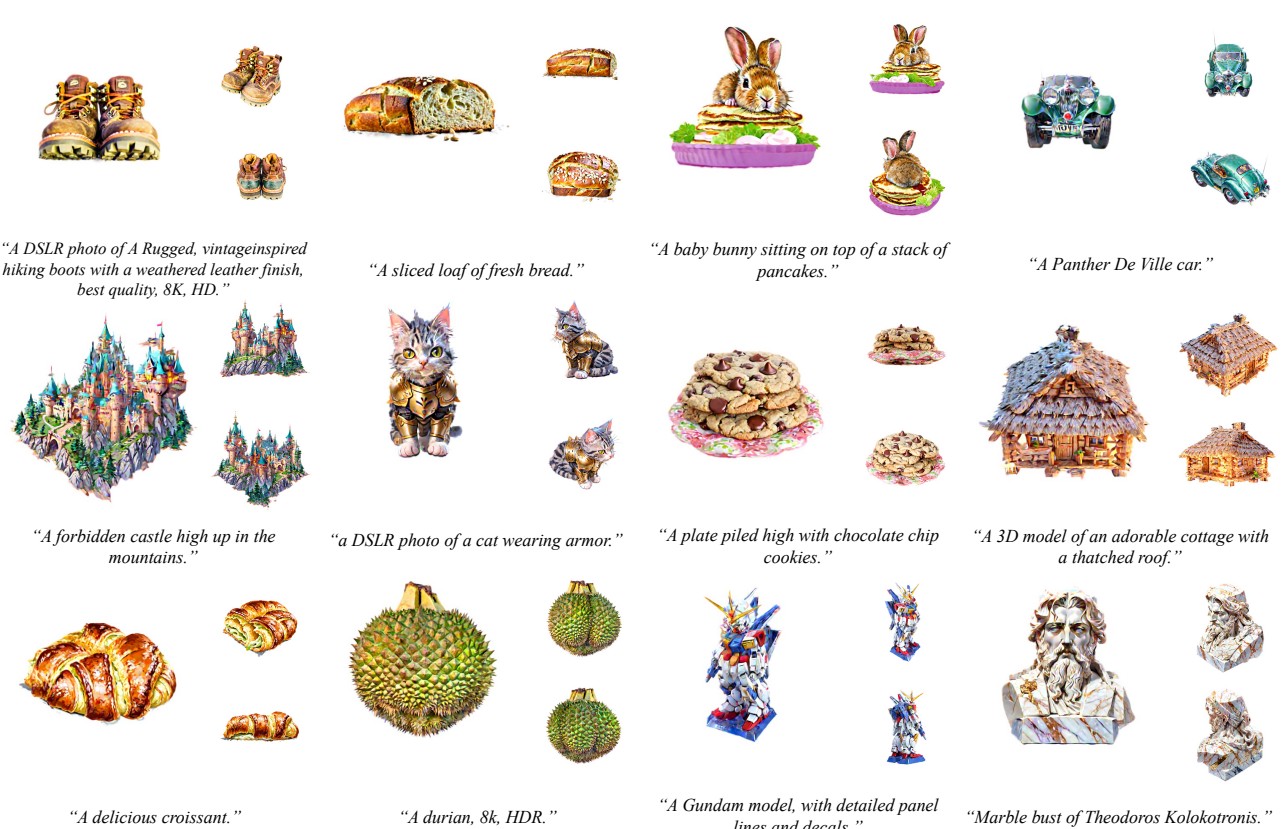

*"A DSLR photo of A Rugged, vintageinspired hiking boots with a weathered leather finish, best quality, 8K, HD."*

*"A sliced loaf of fresh bread."*

*"A baby bunny sitting on top of a stack of pancakes."*

*"A Panther De Ville car."*

*"A forbidden castle high up in the mountains."*

*"a DSLR photo of a cat wearing armor."*

*"A plate piled high with chocolate chip cookies."*

*"A 3D model of an adorable cottage with a thatched roof."*

*"A delicious croissant."*

*"A durian, 8k, HDR."*

*"A Gundam model, with detailed panel lines and decals."*

*"Marble bust of Theodoros Kolokotronis."*

*Figure 14.* **More results generation by our UDS.** Please zoom in for details.

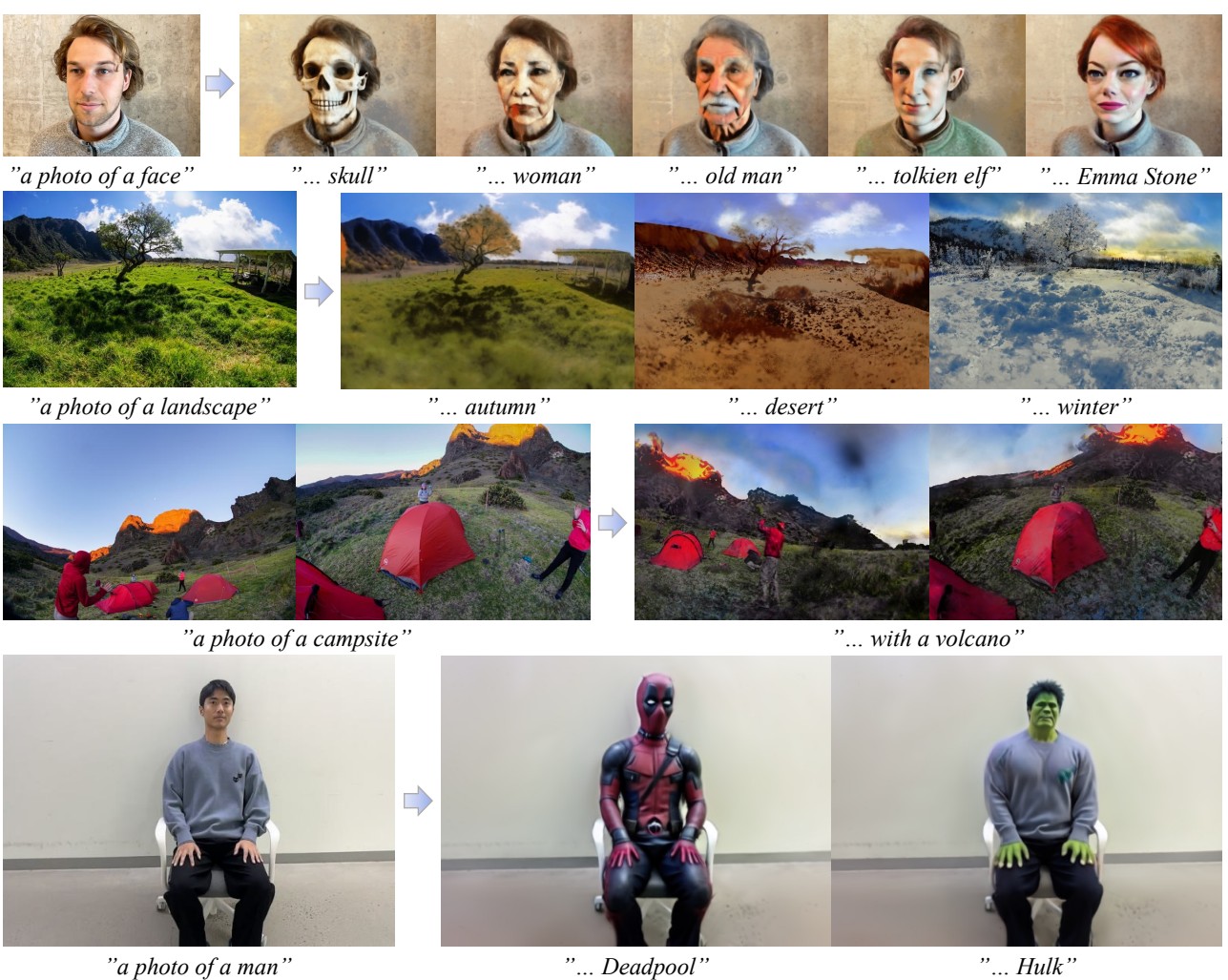

*"a photo of a face"* ”... skull” ”... woman” ”... old man” ”... tolkien elf” ”... Emma Stone”

*"a photo of a landscape"* ”... autumn” ”... desert” ”... winter”

*"a photo of a campsite"* ”... with a volcano”

*"a photo of a man"* ”... Deadpool” ”... Hulk”

*Figure 15.* **More results edited by our UDS.** Please zoom in for details.

