# OpenReview forum: "Rethinking Score Distilling Sampling for 3D Editing and Generation"
_ICML.cc/2025/Conference — ICML 2025 poster_

### Official Review · Reviewer_jT1C · 2025-02-18

**Overall Recommendation:** 2

**Summary:**

This paper introduces a novel variant of SDS, called UDS, which is designed to handle both editing and generation tasks. Via qualitative and quantitative experiments, the authors show that UDS surpasses previous baselines.

## update after rebuttal

As I noted in my official review, I remain unconvinced regarding the paper's contribution.

**Claims And Evidence:**

The main claim of this paper is that UDS enables unified processing of editing and generation tasks. However, the gradients for editing and generation correspond to two distinct variants of SDS (see Equations (14) and (16)). This distinction is even more apparent in Algorithm 1, where an explicit if-else branch separates the two tasks, with only basic time-step sampling and noise sampling shared. The definition of "unify" is somewhat ambiguous. Moreover, the observed performance improvements seem to stem from a more accurate estimation of $x_0$ using DDIM and the introduction of a negative classifier-free guidance term, rather than from unification itself. The authors might consider reframing their contribution as a refinement of the SDS loss, which can be applied to both the editing and generation variants of SDS.

Another claim is that the gradient of UDS is more stable. The authors argue that DDS and PDS rely on a large classifier-free guidance (CFG) scale, leading to instability, but they do not explain why ISM and VSD, which use a small CFG, also exhibit instability. Additionally, the paper lacks a clear justification for why a large CFG may cause instability and how such instability affects the final output. Providing further analysis or empirical evidence would strengthen this claim.

**Essential References Not Discussed:**

In Section 4.2, Line 255, the paper states: *According to some recent work, we know that this approximation is inaccurate and may affect the preservation of identity information in editing tasks", lack of citation*. However, no citation is provided to support this claim.

**Experimental Designs Or Analyses:**

The experiments are not convincing.

1. **Lack of Comparisons with SOTA Methods.** The paper does not compare UDS against SOTA SDS-based methods such as RichDreamer[S1] and Dreamcraft3D[S2].
2. **Insufficient Details in the User Study.** The user study lacks methodological details. conducted to evaluate the overall quality of the generated content lacks detailed methodology. Key aspects such as participant demographics, selection criteria, and statistical significance testing are not provided.
3. **Missing Experimental Details.** Several crucial implementation details are absent, including:
    1. The version of Stable Diffusion used.
    2. The initialization method for 3DGS.
    3. Whether Equation (10) or Equation (15) was used to estimate $x_0$ in UDS
    4. A list of prompts chosen for NeRF-based and 3DGS-based methods.
4. **Limited Comparisons in Figure 3.** Figure 3 only compares LucidDreamer to UDS using two cases, which is insufficient given the similarities between the two methods. A supplementary file with additional comparisons—especially in video format—would provide a more comprehensive evaluation.

[S1] RichDreamer: A Generalizable Normal-Depth Diffusion Model for Detail Richness in Text-to-3D

[S2] Dreamcraft3d: Hierarchical 3d generation with bootstrapped diffusion prior

**Methods And Evaluation Criteria:**

The method appears effective based on the provided results.

**Other Comments Or Suggestions:**

Section 4.2, Line 250 states *Equation (12) is a simplified representation*. But it appears that Equation(14) is the simplified version, while Equation(12) represents the original gradient term of PDS.

**Other Strengths And Weaknesses:**

1. Lack of Discussion on Limitations and Failure Cases
2. Static images are inadequate. Including rotated video of each object would better illustrate the method’s effectiveness.
3. Notable inconsistencies exists in the generated hands (Figure 2, Last Row)

**Questions For Authors:**

See above

**Relation To Broader Scientific Literature:**

The paper builds on prior work in text-to-3D generation, including DreamFusion, Magic3D, and LucidDreamer, while also relating to differentiable 3D representations such as NeRF and 3D Gaussian Splatting.

**Theoretical Claims:**

I have reviewed the proofs, and to the best of my knowledge, they are correct.

---

> ### Author Rebuttal · Authors · 2025-03-28
>
> Thank you for your detailed comments and valuable suggestions, which have greatly improved the quality of our paper. Below, we address your concerns and clarify potential misunderstandings:
>
> >  **Experimental**
>
> **1.** As shown in the [Figure 1](https://anonymous.4open.science/r/15488/1.pdf), we provide a quantitative comparison of our NeRF-based results, namely UDS+DreamFusion, with RichDreamer and Dreamcraft3D.
>
> **2.** Thank you for your suggestion, we will include the following content in the revised version.
>
> We conducted a user study to evaluate the performance of different methods based on human preferences. In the generation task, we showed participants a side-by-side comparison of 3D assets generated by each method. In each trial, participants received a text prompt and a rotated video of multiple candidate 3D assets generated using different methods. In the editing task, we showed the side-by-side effect of each method editing a 3D scene. In each trial, participants received a text prompt, a reference scene, and videos of multiple candidate 3D scenes edited using different methods. [Figure 4](https://anonymous.4open.science/r/15488/4.pdf) is a screenshot of the user study page. We collected responses from a total of 102 participants. Each participant randomly performed 50 to 100 trials, and their selection data were recorded for subsequent analysis. To ensure the diversity and fairness of the evaluation results, the order of presentation of candidate content was randomly arranged in 50 to 100 trials for each participant.
>
> **3.1** In our work, all methods use SD 2.1 for the generation task and SD 1.5 for the editing task.
>
> **3.2** We follow LucidDreamer to use Pointe to initialize 3D GS.
>
> **3.3** We have conducted relevant ablation experiments on these two estimation methods (Tweedie formula and DDIM multi-step denoising) in the main text.
>
> **3.4** Regarding the prompts, as described in the main text, for NeRF-based methods, we use 15 prompts from Magic3D and 415 prompts from DreamFusion (these prompts can be easily found on their official website or threestudio). For 3D GS-based methods, we will add a prompt list in the revised version. We have already shown about 30 prompts in the main text.
>
> **4.** Due to ICML policy, we can't seem to provide videos. As shown in [Figure 5](https://anonymous.4open.science/r/15488/5.pdf), we provide more comparisons with multiple views of LucidDreamer.  For fair comparison, we use the same hyperparameters.
>
> >  **References**
>
> We will add citations to the revised version, e.g. DPS [1].
>
> [1] Diffusion Posterior Sampling for General Noisy Inverse Problems. ICLR (2023)
>
> >  **Weaknesses**
>
> **1.** In generation tasks, success depends on the initialization; if the initialization is poor, the generation is likely to fail. In editing tasks, when there is a large gap between the prompt and the original scene, failure may occur. For example, when changing the prompt from "A photo of a plant" to "a photo of balls", the result often does not generate balls on the branches, but rather places them at random positions in the scene. We will include the above content in the revised version.
>
> **2.** Please refer to **Experimental 4**.
>
> **3.** Our work focuses on exploring a natural and unified formulation of generation and editing tasks rather than on solving the consistency issue. We believe that the inconsistencies in the generated hands can be easily addressed by incorporating multi-view information and geometric constraints. However, this is not the primary focus of our work.
>
> >  **Other Comments**
>
>
> Please note that Equation 12 actually contains some important coefficients of PDS, which we simplify in the main text for easier understanding, while Equation 14 is our proposed method.

---

> > ### Comment · Reviewer_jT1C · 2025-04-03
> >
> > I appreciate the authors' efforts in providing the rebuttal, however most of my concerns are not addressed.
> >
> > My concerns with the contribution of the unifying framework remain unaddressed. Although the idea of a unified approach is interesting, the practical benefits is unclear. Eq. 14 and 16 indicate that the editing and generation process are trained with seperate SDS losses. In this case, what is the performance gain from using a shared architecture for editing and generation? The effectiveness of this unified framework remains ambiguous and therefore reducing the potential contribution to the community.
> >
> > The experimental results are also worrying weak.
> >
> > - The generation results in Figure 1 do not show significant improments over existing methods. For example, the cookie case shows unnatural geometry (some cookies extend to near plane and blend with the distorted plate). In addition, more qualitative results or cases from existing methods should be presented for a thorough evaluation. 4 cases are insufficient to tell the general robustness of the proposed method, for example, the proposed method could be overfitted to these cases.
> > - The editing results also show inconsistency, e.g. different hand colors in the batman case. As much as the authors claim that consistency is not their research goal, it remains a crucial criteria in 3D editing and failing to resolve this would hamper the practical usefulness of the method.
> > - SDS-based methods are known for requiring hours to optimize one scene. If the proposed method is competitive in this aspect, the authors should demonstrate this by comparing the efficiency with more recent state-of-the-art methods, i.e. Trellis, InstantMesh, 3DTopiaXL. Otherwise, with a much longer optimization time, the impact and practical application of the proposed method would be very limited.
> >
> > I have also read other reviewers' feedback and the authors' responses. I agree with reviewer `u3Tp` that the performance gain over existing methods are not obvious, and if Trellis is too hard to compete with, the authors should compare with other SOTA methods as mentioned above. In the authors' response, reporting CLIP score only on front view cannot give reliable results on the text-3D alignment on surrounding views. The authors should follow convention practice (i.e. as in Trellis) to provide a multi-view CLIP score, PSNR, 3D-aware metric (e.g. T$^3$Bench, GPTEval3D) and user study.
> >
> > To summarize, given the ambiguous contribution and weak evaluation results, I think the paper in its current state is below the bar of ICML and I decide to vote for rejection. That said, I remain open to discussion should other reviewers hold different views.

---

> > > ### Author Response · Authors · 2025-04-03
> > >
> > > 1. **Please note that the core of our work is to explore "whether a unified gradient formulation theoretically exists that can handle both editing and generation tasks simultaneously". But in engineering practice, editing and generation differ greatly, and generation tasks do not require a reference scene, which is why we have always emphasized that the theoretical formulation is unified (Similar to some 2D unified methods, they use the same features in the same framework, but they still have different task heads).** Each part of our methodology revolves around this central argument (see Sec. 4). As reviewers **u3Tp** and **uKsu** pointed out, we revisit the theory of probability density distillation in the editing and generation tasks in our work, revealing the internal connection between the two (see Sec. 4.1), which is an important part of our work.
> > >
> > > 2. **Our experimental design aims to verify the feasibility of this theoretically unified formulation rather than simply achieving a clear numerical advantage over existing methods.** You mentioned that the generation results do not show significant improvement compared to existing **feed-forward methods** (It is worth noting that we have different types of methods.), which actually reinforces our argument that the unified gradient formulation can offer a new theoretical perspective while ensuring performance comparable to current SOTA methods. In addition, it is important to note that the results presented in our rebuttal are based on DreamFusion+UDS (with only a single stage of optimization), whereas the RichDreamer and Dreamcraft3D methods you mentioned use three stages of optimization, resulting in noticeable improvements in geometry and texture. **(It is impossible to complete a comprehensive evaluation in the rebuttal stage. Each of these two methods requires more than 4 hours of optimization for a single scene.)**
> > >
> > > 3. **Regarding the editing task, as explained in our work, our focus is on explaining how UDS works effectively in both tasks rather than developing an editing strategy that completely outperforms existing methods.** On the other hand, our UDS achieves a significantly faster optimization speed and higher quality in editing tasks compared to PDS and DDS, and it shows better stability (in the same scene, PDS and DDS sometimes fail to well edit, as noted by reviewer **uKsu** in the “bamboo” and “apple” scenes). Starting from the gradient terms of PDS and DDS, we demonstrate that editing and generation can be theoretically summarized into the same gradient formulation. **Please note that this interpretation is entirely novel.**
> > >
> > > 4. **We firmly believe that understanding why current score distillation methods work is very important.** As acknowledged by reviewers **pueN**, **tQkB**, and **oCeh**, our new insights into score distillation and our experimental validation offer a fresh perspective to the field. Given that many studies focus mainly on single-task optimization, our work offers a unique approach to unifying editing and generation.
> > >
> > > 5. As is well known, there are clear differences in the basic principles between SDS-based methods and feed-forward methods, and so far no method has been able to compare them within the same framework. Moreover, Trellis and 3DTopiaXL just have been accepted at CVPR 2025, while the submission deadline for ICML 2025 is January 30, 2025, and the notification date for CVPR 2025 is February 26, 2025. As reviewer **u3Tp** pointed out, since Trellis was just released and the rebuttal period was limited, we have not yet found an evaluation metric that can be used for both completely different types of methods. Even if you use GPTEval3D, the comparison is not fair due to the different rendering angles and number of views. Therefore, we simply compare the feed-forward method. It should be noted that our method is clearly superior to early feed-forward methods, and the training datasets of these feed-forward methods already cover most of the 3D data.
> > >
> > > 6. **We sincerely hope that the primary focus of our work, which is to present essential insights and arguments, aligns with your considerations. Our goal is to contribute to a better understanding of the field rather than merely developing a method that outperforms previous ones.**

---

### Official Review · Reviewer_u3Tp · 2025-03-13

**Overall Recommendation:** 3

**Summary:**

This paper tackles the problem of 3D asset generation and editing using SDS. The paper identify limitations in existing SDS variants especially for edting. The core contribution is a new method called Unified Distillation Sampling UDS. The key ideas behind UDS are: gradient term unification. A few other improvements like integration with classifier free guidance are also proposed.
The paper demonstrates its effectiveness through experiments on both 3D generation and editing tasks. It compares UDS against several baselines using quantitative metrics and qualitative visual results. Experiments are conducted on NeRF and 3D Gaussian Splatting representations, as well as on SVG editing.

I will structure this review using a strengths and weaknesses template to better organize the points.

**Claims And Evidence:**

see others

**Essential References Not Discussed:**

no

**Experimental Designs Or Analyses:**

see others

**Methods And Evaluation Criteria:**

see others

**Other Comments Or Suggestions:**

no

**Other Strengths And Weaknesses:**

1: The main strength is the proposed unification of 3D generation and editing within a single framework. This is a good contribution, as prior work often treated these as separate problems. The analysis of the gradient terms is  well-motivated.

2: The paper demonstrates that UDS outperforms existing methods in editing tasks, particularly in preserving the identity and details of the original 3D asset while incorporating the desired edits. The visual results are compelling, and the quantitative results support this claim. The authors evaluate UDS on a wide range of tasks (NeRF generation/editing, 3D GS generation, SVG editing) and compare it against a comprehensive set of baselines. They include both quantitative metrics and user studies, providing a thorough evaluation.

3: The paper provides a solid theoretical justification for UDS, deriving it from an analysis of existing methods. The connection to DDIM inversion for the reconstruction term is a clever idea.

Weaknesses:

1: While UDS achieves better results, the paper acknowledges that using DDIM inversion for the reconstruction term increases computational cost compared to simpler approximations. A more detailed discussion of the trade-off between quality and efficiency would be beneficial. It is unclear how much longer UDS takes compared to, e.g. PDS or DDS. Though in text-3D experiments USD seems to be fast when compared to other baselines.

2: Performance gain seems to be not very significant in both editing and generation tasks.  For generation, can the performance be compared to feed-forward models like trellis (Structured 3D Latents for Scalable and Versatile 3D Generation) . trellis might be very hard to compete with (and its recently published) so maybe testing a few weaker feed-forward 3D gen models would also be beneficial.

3: The user study gives little information. More detail, such as the number of participants, the number of samples presented to the participants, and the interface used, should be included in the appendix.

**Questions For Authors:**

Are there specific types of edits or generation tasks where UDS might struggle? Are there failure cases to be aware of?

UDS, like other SDS-based methods, relies on the quality and biases of the pre-trained 2D diffusion model. The paper doesn't address how the limitations of the 2D model might propagate to the 3D results. Will UDS be better than SDS in terms of this?

How robust is the method to variations in w, the time step interval (c), and other parameters? Any results or hyper-parameter analysis?

**Relation To Broader Scientific Literature:**

yes, related.

**Theoretical Claims:**

didn't check carefully

---

> ### Author Rebuttal · Authors · 2025-03-28
>
> Thank you for your detailed comments and valuable suggestions, which have greatly improved the quality of our paper. Below, we address your concerns and clarify potential misunderstandings:
>
> >  **Weaknesses**
>
> **1.** We present a comparison of the optimization time of our method with that of PDS and DDS in the editing task. Please note that when using the same training steps, our method performs similarly to PDS and DDS; however, we found that the UDS method converges faster, requiring only one-third of the steps needed by PDS and DDS.
>
>
> | Method | Time |
> | ----------- | ----------- |
> | PDS | ~17 hour 20 mins |
> | DDS | ~17 hour 20 mins |
> | Tweedie (Our) Full steps | ~17 hour 20 mins |
> | Tweedie (Our) 1/3 steps | ~5 hour 42 mins |
> | DDIM (400 steps) (Our) 1/3 steps | ~8 hour 20 mins |
> | DDIM (200 steps) (Our) 1/3 steps | ~10 hour 37 mins |
> | DDIM (100 steps) (Our) 1/3 steps | ~17 hour 15 mins  |
> | DDIM (50 steps) (Our) 1/3 steps | ~49 hour 30 mins |
>
> **2.** We compare several feed-forward methods: the early ShapE, 3DTopia, and the latest TRELLIS. We use the 43 prompts mentioned in our paper for generation. Because it is difficult to find a metric that works well for these two kind of methods, we render only the front view and calculate the CLIP score for evaluation.
>
> | Method | CLIP Score |
> | ----------- | ----------- |
> | ShapE [1] | 0.2134 |
> | 3DTopia [2] | 0.2248 |
> | TRELLIS [3] |  0.2697 |
> | Ours (3D GS) | 0.2531 |
>
> [1] Shap-E: Generating Conditional 3D Implicit Functions
>
> [2] 3DTopia: Large Text-to-3D Generation Model with Hybrid Diffusion Priors
>
> [3] Structured 3D Latents for Scalable and Versatile 3D Generation
>
> **3.** Thank you for your suggestion, we will include the following content in the revised version.
>
> We conducted a user study to evaluate the performance of different methods based on human preferences. In the generation task, we showed participants a side-by-side comparison of 3D assets generated by each method. In each trial, participants received a text prompt and a rotated video of multiple candidate 3D assets generated using different methods. In the editing task, we showed the side-by-side effect of each method editing a 3D scene. In each trial, participants received a text prompt, a reference scene, and videos of multiple candidate 3D scenes edited using different methods. [Figure 4](https://anonymous.4open.science/r/15488/4.pdf) is a screenshot of the user study page. We collected responses from a total of 102 participants. Each participant randomly performed 50 to 100 trials, and their selection data were recorded for subsequent analysis. To ensure the diversity and fairness of the evaluation results, the order of presentation of candidate content was randomly arranged in 50 to 100 trials for each participant.
>
> >  **Questions**
>
> **1.** In generation tasks, success depends on the initialization; if the initialization is poor, the generation is likely to fail. In editing tasks, when there is a large gap between the prompt and the original scene, failure may occur. For example, when changing the prompt from "A photo of a plant" to "a photo of balls", the result often does not generate balls on the branches, but rather places them at random positions in the scene. We will include the above content in the revised version.
>
> **2.** We thank you for raising this question. In fact, SDS-based methods all rely on pre-trained 2D diffusion models, but our UDS makes improvements in gradient construction. Specifically, we use latent embedding $x_0$ to replace $\epsilon$ of previous SDS-based methods. Compared with $\epsilon$, $x_0$ is closer to the data and has less noise, thereby weakening the propagation of 2D model deviations in 3D results. In other words, the unified gradient signal we obtain is more stable, which has been reflected in the experiments that the generated and edited results are richer in details and less distorted.
>
> **3.** We provide ablation study on the time interval $c$ here ([Figure 2. Ablation on different time interval](https://anonymous.4open.science/r/15488/2.pdf)). We provide ablation study on CFG $w$ here ([Figure 3. Ablation on different CFG](https://anonymous.4open.science/r/15488/3.pdf)).

---

> > ### Comment · Reviewer_u3Tp · 2025-04-02
> >
> > Many thanks for providing a well-prepared rebuttal, which does solve most of my concerns. I am inclined to keep my original rating as weak accept.
> >
> > One follow-up discussion, but this is more like an open discussion for the field, so I won't degrade my rating even if there is no reply. TRELLIS seems to have stronger performance than the proposed model, and it also runs much faster. Do you think the gap can be closed by using stronger 2D diffusion models? Are there any specific advantages for SDS-based optimization models compared to those feed-forward ones?

---

> > > ### Author Response · Authors · 2025-04-02
> > >
> > > Thank you for your reply. As is well known, acquiring 3D data is very costly, so we believe that future 3D generation models will mainly adopt a feedforward architecture. In contrast, SDS-based methods will serve more as data generation engines. Although more powerful 2D diffusion models can improve the quality of SDS-generated results to some extent, in the long run, training with large-scale synthetic data is key to further enhancing the performance of feedforward models. This trend has already been observed in the field of 2D vision. For example, the recently proposed “Depth Anything” model is trained entirely on synthetic data and still performs well on monocular depth estimation tasks.

---

### Official Review · Reviewer_tQkB · 2025-03-13

**Overall Recommendation:** 3

**Summary:**

This paper focuses on score-sampling based text-to-3D generation and editing. The authors first analyze the variants of SDS (e.g., DDS and PDS) and identify significant commonalities in their gradient optimization processes. The authors further introduce a unified distillation sampling (UDS) that enables unified processing of text-to-3D generation and editing. Experiments on 3D editing and generation validate the superiority of the proposed UDS.

**Claims And Evidence:**

All claims are supported by theoretical analysis and experiments.

**Essential References Not Discussed:**

Some most recent SDS/DDS variants for text-to-3D generation and editing are missing, which are highly related to this work and should be compared. For instance, NFSD [a], ODS [b] and ISD [c]  are introduced for text-driven 3D generation or editing.

[a] Noise-Free Score distillation, ICLR'24

[b] GG-Editor: Locally Editing 3D Avatars with Multimodal Large Language Model Guidance, MM'24

[c] VividDreamer: Invariant Score Distillation for Hyper-Realistic Text-to-3D Generation, ECCV'24

**Experimental Designs Or Analyses:**

Each ablation experiment is presented with qualitative results using only one example. It would be more convincing to provide some quantitative results or more visual examples.

**Methods And Evaluation Criteria:**

The proposed UDS seamlessly integrates 3D generation and editing. However, the benefits and potential applications of unifying them are unclear. How does the introduced UDS improve generation/editing quality compared to previous approaches?

**Other Comments Or Suggestions:**

N/A

**Other Strengths And Weaknesses:**

- The overall paper is well organized and written.

- Comprehensive experiments are provided covering text-to-3D generation and editing as well as SVG editing.

**Questions For Authors:**

N/A

**Relation To Broader Scientific Literature:**

SDS is one of the most important techniques in text-to-3D generation. Based on SDS, several variants are proposed for image editing (DDS) and text-to-3D editing. This paper analyzes recent variants of SDS (i.e., DDS and PDS) and introduces a unified UDS for text-to-3D generation and editing.

**Theoretical Claims:**

I am wondering why the rightmost term is independent to the identity preservation term in Equation (9)?

---

> ### Author Rebuttal · Authors · 2025-03-27
>
> Thank you for your detailed comments and valuable suggestions, which have greatly improved the quality of our paper. Below, we address your concerns and clarify potential misunderstandings:
>
> >  **Methods And Evaluation Criteria**
>
> Most previous works have focused on a single 3D generation or editing task. Despite the natural and widely recognized formal similarities between editing and generation, no work has explored the two jointly. Our work attempts to investigate both tasks simultaneously and successfully uses the same framework to complete different tasks, which provides a theoretical and experimental basis for subsequent research based on SDS methods. In the generation task, most previous methods use $\epsilon$-guided reconstruction terms for optimization, while our approach uses latent embeddings $x_0$ that are closer to the real data. Since $x_0$ has less noise, this makes the generation process more stable and the generation quality is improved. For the editing task, we prove through derivation that compared to PDS, we omit the implicit identity constraint term in the gradient. This enables editing to be performed in the regular CFG setting and accelerates convergence, thereby improving the editing effect.
>
> >  **Theoretical**
>
> The term on the right corresponds to the gradient expression used in DDS, specifically,
> $\boldsymbol{\epsilon} _ \theta (\boldsymbol{x} _ t^{\text{tgt}}, y^{\text{tgt}}, t) - \boldsymbol{\epsilon} _ \theta (\boldsymbol{x} _ t^{\text{src}}, y^{\text{src}}, t)$.
> The estimated noise of $\boldsymbol{x}_t^{\text{src}}$serves as a reference for preserving the identity of the source. Although DDS achieves strong editing results in 2D image applications, it suffers from quality degradation when applied to 3D editing. This degradation occurs because 3D editing requires stronger identity preservation than 2D editing. In DDS, the optimization process minimizes the difference between the SDS losses of the source and target, but it does not introduce any additional constraint to explicitly preserve identity. As a result, the lack of regularization allows the optimization to drift away from the source identity. In other words, the gradient term of DDS is not independent of the identity preserving term, but lacks an explicit identity preserving constraint.
>
> >  **Experimental**
>
> We provide additional quantitative ablation experiments. Due to rebuttal period constraints, for the 3D GS-based generation tasks, we conducted ablation on only 20 out of the 43 prompts available in 3D GS. For the NeRF-based generation tasks, ablation experiments were performed on 15 prompts from Magic3D. For the NeRF editing tasks, we conducted ablation on 10 different prompts in the ‘man’ scene.
>
> **Table1. Ablation for different approximate methods. (NeRF-based editing)**
>
> | Method | CLIP Score |
> | ----------- | ----------- |
> | DDIM Inverse (200 steps) | 0.2595 |
> | Tweedie's Formula | 0.2544 |
>
> **Table2. Ablation for SDS and UDS with different generation frameworks. (NeRF-based generation)**
>
> | Method | CLIP Score |
> | ----------- | ----------- |
> | DreamFusion (SDS CFG=100) | 0.2616 |
> | DreamFusion (UDS CFG=7.5) | 0.2874 |
> | Fantasia3D (SDS CFG=100) | 0.2730 |
> | Fantasia3D (UDS CFG=7.5) | 0.2959 |
>
> **Table3. Ablation for generation task. Add noise by DDIM inverse strategy. (3D GS-based generation)**
>
>
> | Method | CLIP Score |
> | ----------- | ----------- |
> | W/o DDIM Inverse | 0.2863 |
> | W/ DDIM Inverse | 0.2977 |
>
>
> >  **References**
>
> Thank you very much for recognizing our work. We will discuss NFSD, ODS, and ISD in the revised version.

---

> > ### Comment · Reviewer_tQkB · 2025-04-05
> >
> > Thanks for your response. The response has addressed most of my concerns.
> >
> > I am wondering the potential application of the proposed unified generation and editing framework?
> >
> > Overall, I would like to keep my positive rating, and I hope the revision can be included in the final version.

---

> > > ### Author Response · Authors · 2025-04-07
> > >
> > > Thank you for your response. Regarding the potential application of our proposed unified generation and editing framework, we believe it can act as a key data generation engine for future 3D generation models. As well all know, obtaining real 3D data is very costly, so feedforward models that rely on large-scale synthetic data will likely become the norm. Our framework can generate detailed 3D assets and also edit existing ones with high fidelity, which makes it well suited for creating synthetic datasets for training feedforward architectures.

---

### Official Review · Reviewer_uKsu · 2025-03-13

**Overall Recommendation:** 3

**Summary:**

This paper points out that previous SDS and its variants have only performed well in generation or editing tasks. To address this, a unified approach integrating 3D asset generation and editing is proposed. The authors observe that the generation and editing processes in SDS and its variants share a common fundamental gradient term, leading to the introduction of UDS, which enables a unified approach to handling both 3D editing and generation tasks. Extensive experiments validate its effectiveness across various applications, demonstrating the capability of UDS in unifying 3D editing and generation tasks.

**Claims And Evidence:**

The authors argue that DDS fails in 3D editing while PDS remains effective due to PDS providing an explicit identity preservation term. This claim is reasonable, as it represents the most significant difference between PDS and DDS. The authors support this argument with theoretical analysis and mathematical formulations.

Furthermore, the authors propose a simplified version of PDS by retaining the identity preservation term and the classifier-free guidance (CFG) term while removing the \delta^{recon}_{x_t} term to better control the editing process. This claim is also justified, as demonstrated through comparative experiments with PDS. Specifically, in the example of transforming “a photo of a bamboo” into “a photo of an apple tree,” PDS produces an unreasonable result by incorporating the Apple company logo, whereas the proposed method achieves a more appropriate editing outcome.

**Essential References Not Discussed:**

This paper focuses on 3D editing tasks but lacks citations of relevant literature on 3D editing for 3D Gaussian Splatting (3DGS), such as GaussianEditor [1] and DYG [2].

[1] Gaussianeditor: Editing 3d gaussians delicately with text instructions
[2] Drag Your Gaussian: Effective Drag-Based Editing with Score Distillation for 3D Gaussian Splatting

**Experimental Designs Or Analyses:**

The experimental design and analysis in this paper are reasonable and effective. However, since this work is similar to ISM and introduces the hyperparameter time interval c in the generation task, I believe an ablation study on this hyperparameter is necessary.

**Methods And Evaluation Criteria:**

The experimental setup, comparisons with other methods, and evaluation criteria in this paper are reasonable and effective.

**Other Comments Or Suggestions:**

Reference to "Other Strengths And Weaknesses".

**Other Strengths And Weaknesses:**

In equation (4), the symbol θ is used to denote the optimizable parameters of the 3D scene, but it is also used to represent the diffusion model parameter ε_θ. Referring to equation (2), where φ is used for the diffusion model parameter, I believe this is likely a typographical error. The same issue appears in multiple equations throughout the paper.

In equation (5), delta^{cls}_{x_t} includes the CFG weight w. However, referring to equations (11), (12), and (13), where w is multiplied by delta^{cls}, I suspect that delta^{cls}_{x_t} in equation (5) should not include the CFG weight w.

In line 206, the t in delta^{recon}_{xt} should be a subscript, but it appears to be a typographical error.

**Questions For Authors:**

This paper compares the effects of the Tweedie formula and multi-step DDIM Inversion through ablation experiments. Since multi-step DDIM Inversion incurs significant time overhead, I would like to understand the exact difference in runtime between the two methods.

I would like to gain a deeper understanding of the design rationale behind equation (17). Specifically, why was the time interval c introduced, and how does its value impact the results?

**Relation To Broader Scientific Literature:**

Previous works have often focused on either a single 3D generation task or a 3D editing task. For example, DreamFusion specializes in 3D generation, while PDS focuses on 3D editing, lacking a unified framework. Given the similarities between generation and editing tasks, this paper proposes UDS to integrate both into a unified approach.

**Theoretical Claims:**

I have not verified all of the correctness of the theoretical claims and proofs.

---

> ### Author Rebuttal · Authors · 2025-03-28
>
> Thank you for your detailed comments and valuable suggestions, which have greatly improved the quality of our paper. Below, we address your concerns and clarify potential misunderstandings:
>
> >  **References**
>
> We will include the **Gaussianeditor** and **Drag Your Gaussian** discussions in the revised version.
>
> >  **Weaknesses**
>
> Thank you for your careful review of our symbols. We will correct these errors in the revised version as follows:
>
> - $\epsilon_\theta$ -> $\epsilon_\phi$.
> - $\underbrace{w(\boldsymbol{\epsilon} _ \theta (\boldsymbol{x}_t,t,y) - \boldsymbol{\epsilon} _ \theta (\boldsymbol{x}_t,t,\emptyset))} _ {\delta _ {\boldsymbol{x}_t}^\text{cls}}$ -> $w\underbrace{(\boldsymbol{\epsilon} _ \theta (\boldsymbol{x}_t,t,y) - \boldsymbol{\epsilon} _ \theta (\boldsymbol{x}_t,t,\emptyset))} _ {\delta _ {\boldsymbol{x}_t}^\text{cls}}$
> - $\delta_{\boldsymbol{x}t}^\text{recon}$ -> $\delta_{\boldsymbol{x}_t}^\text{recon}$
>
> >  **Questions**
>
> **1.** We provide the training times for the generation and editing tasks using the Tweedie formula and multi-step DDIM Inversion. Please note that we found that in the generation task, the DDIM interval does not significantly affect the quality of the results, and the differences in the 3D assets produced by DDIM and the Tweedie formula are minimal. Therefore, to speed up the generation process, we directly use the Tweedie formula for the generation task. For the editing task, we found that DDIM affects the editing outcome; a shorter DDIM interval retains more of the original information, but it also increases the editing time. (We also conducted corresponding ablation experiments in Figure 4 in the main text.)
>
> **Table 1. Training time of DDIM (different time intervals) and Tweedie in the generation task. Note that all experiments are performed on a 3090.**
>
> | Method | Time |
> | ----------- | ----------- |
> | Tweedie | ~52 mins |
> | DDIM (400 steps) | ~1 hour 21 mins |
> | DDIM (200 steps) | ~1 hour 30 mins |
> | DDIM (100 steps) | ~1 hour 49 mins  |
> | DDIM (50 steps) | ~3 hour 17 mins |
>
> **Table 2. Training time of DDIM (different time intervals) and Tweedie in the edit task. Note that all experiments are performed on a A6000.**
>
>
> | Method | Time |
> | ----------- | ----------- |
> | Tweedie | ~5 hour 5 mins |
> | DDIM (400 steps) | ~8 hour 20 mins |
> | DDIM (200 steps) | ~10 hour 37 mins |
> | DDIM (100 steps) | ~17 hour 15 mins  |
> | DDIM (50 steps) | ~49 hour 30 mins |
>
> **2.** The time interval $c$ is inspired by recent approaches that use DDIM-based noise addition, such as ISM. In ISM, the gradient is guided by an $\epsilon$ that is highly correlated with the data at two time steps; in contrast, our UDS guides the gradient using a latent embedding $x_0$ that is closer to the data at two time steps. We provide an ablation study on the time interval $c$ here ([Figure 2. Ablation on different time interval](https://anonymous.4open.science/r/15488/2.pdf)).

---

### Decision · Program_Chairs · 2025-05-01

**Decision:**

Accept (poster)

**Comment:**

Based on thorough and well-motivated analyses of SDS, DDS, and PDS, the authors formulated the unified approach toward text-to-3D generation and editing. Experiment and analysis are well-excuted and clearly conveyed. A major part of reviewers agreed that this work deserves acceptance after author rebuttal. The theoretical unficiation of DDS and PDS, leading to the proposed UDS is appealing and well-motivated. AC agreed with most of the reviewers' positive opinions, evaluating that they outweigh the minor weaknesses.